# *DeepPipe*: DEEP, MODULAR AND EXTENDABLE REPRESENTATIONS OF MACHINE LEARNING PIPELINES

## ABSTRACT

Finding accurate Machine Learning pipelines is essential in achieving state-of-the-art AI predictive performance. Unfortunately, most existing Pipeline Optimization techniques rely on flavors of Bayesian Optimization that do not explore the deep interaction between pipeline stages/components (e.g. between hyperparameters of the deployed preprocessing algorithm and the hyperparameters of a classifier). In this paper, we are the first to capture the deep interaction between components of a Machine Learning pipeline. We propose embedding pipelines in a deep latent representation through a novel per-component encoder mechanism. Such pipeline embeddings are used with deep kernel Gaussian Process surrogates inside a Bayesian Optimization setup. Through extensive experiments on three large-scale meta-datasets, including Deep Learning pipelines for computer vision, we demonstrate that learning pipeline embeddings achieves state-of-the-art results in Pipeline Optimization.

## 1 INTRODUCTION

Machine Learning (ML) has proven to be successful in a wide range of tasks such as image classification, natural language processing, and time series forecasting. In a supervised learning setup, practitioners need to define a sequence of stages comprising algorithms that transform the data (e.g. imputation, scaling) and produce an estimation (e.g. through a classifier or regressor). The selection of the algorithms and their hyperparameters, known as Pipeline Optimization (Olson & Moore, 2016) or pipeline synthesis (Liu et al., 2020; Drori et al., 2021) is challenging. Firstly, the search space contains conditional hyperparameters, as only some of them are active depending on the selected algorithms. Secondly, this space is arguably bigger than the one for a single algorithm. Therefore, previous work demonstrates how this pipeline search can be automatized and achieve competitive results (Feurer et al., 2015; Olson & Moore, 2016). Some of these approaches include Evolutionary Algorithms (Olson & Moore, 2016), Reinforcement Learning (Rakotoarison et al., 2019; Drori et al., 2021) or Bayesian Optimization (Feurer et al., 2015; Thornton et al., 2012; Alaa & van der Schaar, 2018).

Pipeline Optimization (PO) techniques need to capture the complex interaction between the algorithms of a Machine Learning pipeline and their hyperparameter configurations. Unfortunately, no prior method uses Deep Learning to encapsulate the interaction between pipeline components. Prior work trains performance predictors (a.k.a. surrogates) on the concatenated hyperparameter space of all algorithms (raw search space), for instance, using random forests (Feurer et al., 2015) or finding groups of hyperparameters to use on kernels with additive structure (Alaa & van der Schaar, 2018). On the other hand, transfer learning has been shown to decisively improve PO by transferring efficient pipelines evaluated on other datasets (Fusi et al., 2018; Yang et al., 2019; 2020). Our method is the first to introduce a deep pipeline representation that is meta-learned to achieve state-of-the-art results in terms of the quality of the discovered pipelines.

We introduce *DeepPipe*, a neural network architecture for embedding pipeline configurations on a latent space. Such deep representations are combined with Gaussian Processes (GP) for tuning pipelines with Bayesian Optimization (BO). We exploit the knowledge of the hierarchical search space of pipelines by mapping the hyperparameters of every algorithm through per-algorithm encoders to a hidden representation, followed by a Fully Connected Network that receives the concatenated representations as input. Additionally, we show that meta-learning this network through evaluations

on auxiliary tasks improves the quality of BO. Experiments on three large-scale meta-datasets show that our method achieves the new state-of-the-art in Pipeline Optimization.

Our contributions are as follows:

- We introduce *DeepPipe*, a surrogate for BO that achieves peak performance when optimizing a pipeline for a new dataset through transfer learning.
- We present a novel and modular architecture that applies different encoders per stage and yields better generalization in low meta-data regimes, i.e. few/no auxiliary tasks.
- We conduct extensive evaluations against seven baselines on three large meta-datasets, and we further compare against rival methods in OpenML datasets to assess their performances under time constraints.
- We demonstrate that our pipeline representation helps achieving state-of-the-art results in optimizing pipelines for fine-tuning deep computer vision networks.

## 2   RELATED WORK

**Full Model Selection (FMS)** is also referred to as Combined Algorithm Selection and Hyperparameter optimization (CASH) (Hutter et al., 2019; Feurer et al., 2015). FMS aims to find the best model and its respective hyperparameter configuration (Hutter et al., 2019). A common approach is to use Bayesian Optimization with surrogates that can handle conditional hyperparameters, such as Random Forest (Feurer et al., 2015), tree-structured Parzen estimators (Thornton et al., 2012), or ensembles of neural networks (Schilling et al., 2015).

**Pipeline Optimization (PO)** is a generalization of FMS where the goal is to find the algorithms and their hyperparameters for different stages of a Machine Learning Pipeline. Common approaches model the search space as a tree structure and use reinforcement learning (Rakotoarison et al., 2019; Drori et al., 2021; de Sá et al., 2017), evolutionary algorithms (Olson & Moore, 2016), Hierarchical Task Networks (Mohr et al., 2018) for searching pipelines. Other approaches use Multi-Armed Bandit strategies to optimize the pipeline, and combine them with Bayesian Optimization (Swearingen et al., 2017) or multi-fidelity optimization (Kishimoto et al., 2021). Alaa & van der Schaar (2018) use additive kernels on a Gaussian Process surrogate to search pipelines with BO that groups the algorithms in clusters and fit their hyperparameters on independent Gaussian Processes, achieving an effectively lower dimensionality per input. By formulating the Pipeline Optimization as a constrained optimization problem, Liu et al (Liu et al., 2020) introduce a method based on alternating direction method of multipliers (ADMM) (Gabay & Mercier, 1976).

**Transfer-learning for Pipeline Optimization and CASH** leverages information from previous (auxiliary) task evaluations. Few approaches use dataset meta-features to warm-start BO with good configurations from other datasets (Feurer et al., 2015; Alaa & van der Schaar, 2018). As extracting meta-features demands computational time, follow-up works find a portfolio based on these auxiliary tasks (Feurer et al., 2020). Another popular approach is to use collaborative filtering with a matrix of pipelines vs task evaluations to learn latent embeddings of pipelines. OBOE obtains the embeddings by applying a QR decomposition of the matrix on a time-constrained formulation (Yang et al., 2019). By recasting the matrix as a tensor, Tensor-OBOE (Yang et al., 2020) finds latent representations via the Tucker decomposition. Furthermore, Fusi et al. (2018) apply probabilistic matrix factorization for finding the latent pipeline representations. Subsequently, they use the latent representations as inputs for a Gaussian Process, and explore the search space using BO.

## 3   PRELIMINARIES

### 3.1   PIPELINE OPTIMIZATION

The pipeline of a ML system consists of a sequence of $N$ stages (e.g. dimensionality reducer, standardizer, encoder, estimator (Yang et al., 2020)). At each stage $i \in \{1 \ldots N\}$ a pipeline includes one algorithm[1] from a set of $M_i$ choices (e.g. the *estimator* stage can include the algorithms

---

[1]AutoML systems might select multiple algorithms in a stage, however, our solution trivially generalizes by decomposing stages into new sub-stages with only a subset of algorithms.

{SVM, MLP, RF}). Algorithms are tuned through their hyperparameter search spaces, where $\lambda_{i,j}$ denotes the configuration of the $j$-th algorithm in the $i$-th stage. Furthermore, let us denote a pipeline $p$ as the set of indices for the selected algorithm at each stage, i.e. $p := (p_1, \ldots, p_N)$, where $p_i \in \{1 \ldots M_i\}$ represents the index of the selected algorithm at the $i$-th pipeline stage. The hyperparameter configuration of a pipeline is the unified set of the configurations of all the algorithms in a pipeline, concretely $\lambda(p) := (\lambda_{1,p_1}, \ldots, \lambda_{N,p_N}), \lambda_{i,p_i} \in \Lambda_{i,p_i}$. Pipeline Optimization demands finding the optimal pipeline $p^*$ and its optimal configuration $\lambda(p^*)$ by minimizing the validation loss of a trained pipeline on a dataset $\mathcal{D}$ as shown in Equation 1.

$$(p^*, \lambda(p^*)) = \underset{\substack{p \in \{1 \ldots M_1\} \times \cdots \times \{1 \ldots M_N\}, \\ \lambda(p) \in \Lambda_{1,p_1} \times \cdots \times \Lambda_{N,p_N}}}{\arg \min} \mathcal{L}^{\text{val}}(p, \lambda(p), \mathcal{D}) \tag{1}$$

From now we will use the term **pipeline configuration** for the combination of a sequence of algorithms $p$ and their hyperparameter configurations $\lambda(p)$, and denote it simply as $p_\lambda := (p, \lambda(p))$.

## 3.2 BAYESIAN OPTIMIZATION

Bayesian optimization (BO) is a mainstream strategy for optimizing ML pipelines (Feurer et al., 2015; Hutter et al., 2011; Alaa & van der Schaar, 2018; Fusi et al., 2018; Schilling et al., 2015). Let us start with defining a history of $Q$ evaluated pipeline configurations as $\mathcal{H} = \{(p_\lambda^{(1)}, y^{(1)}), \ldots, (p_\lambda^{(Q)}, y^{(Q)})\}$, where $y^{(q)} \sim \mathcal{N}(f(p_\lambda^{(q)}), \sigma_q^2)$ is a probabilistic modeling of the validation loss $f(p_\lambda^{(q)})$ achieved with the $q$-th evaluated pipeline configuration $p_\lambda^{(q)}, q \in \{1 \ldots Q\}$. Such a validation loss is approximated with a surrogate model, which is typically a Gaussian process (GP) regressor. We measure the similarity between pipelines via a kernel function $k : \text{dom}(p_\lambda) \times \text{dom}(p_\lambda) \to \mathbb{R}_{>0}$ parameterized with $\theta$, and represent similarities as a matrix $K'_{q,\ell} := k(p_\lambda^{(q)}, p_\lambda^{(\ell)}; \gamma), K' \in \mathbb{R}_{>0}^{Q \times Q}$. Since we consider noise, we define $K = K' + \sigma_p I$. A GP estimates the validation loss $f_*$ of a new pipeline configuration $p_\lambda^{(*)}$ by computing the posterior mean $\mathbb{E}[f_*]$ and posterior variance $V[f_*]$ as:

$$\mathbb{E}\left[f_* \mid p_\lambda^{(*)}, \mathcal{H}\right] = K_*^T K^{-1} y, \quad V\left[f_* \mid p_\lambda^{(*)}, \mathcal{H}\right] = K_{**} - K_*^T K^{-1} K_* \tag{2}$$

where $K_{*,q} = k\left(p_\lambda^{(*)}, p_\lambda^{(q)}; \gamma\right), K_* \in \mathbb{R}_{>0}^Q$, and $K_{**} = k\left(p_\lambda^{(*)}, p_\lambda^{(*)}; \gamma\right), K_{**} \in \mathbb{R}_{>0}$.

BO is an iterative process that alternates between fitting a GP regressor as described above and selecting the next pipeline configuration to evaluate (Snoek et al., 2012). A description of how BO finds pipelines using a GP surrogate is provided in Appendix I.

## 4 *DeepPipe*: BO WITH DEEP PIPELINE CONFIGURATIONS

To apply BO to Pipeline Optimization (PO) we must define a kernel function that computes the similarity of pipeline configurations, i.e. $k\left(p_\lambda^{(q)}, p_\lambda^{(\ell)}; \theta\right) = ?$. Prior work exploring BO for PO use kernel functions directly on the raw concatenated vector space of selected algorithms and their hyperparameters (Alaa & van der Schaar, 2018) or use surrogates without dedicated kernels for the conditional search space (Feurer et al., 2015; Olson & Moore, 2016; Schilling et al., 2015).

However, we hypothesize that these approaches cannot capture the deep interaction between pipeline stages, between algorithms inside a stage, between algorithms across stages, and between different configurations of these algorithms. In order to address this issue we propose a simple, yet powerful solution to PO: learn a deep embedding of a pipeline configuration and apply BO with a deep kernel (Wistuba & Grabocka, 2021; Wilson et al., 2016).

This is done by *DeepPipe*, which searches pipelines in a latent space using BO with Gaussian Processes. We use a neural network $\phi(p_\lambda; \theta) : \text{dom}(p_\lambda) \to \mathbb{R}^Z$ with weights $\theta$ to project a pipeline configuration to a $Z$-dimensional space. Then, we measure the pipelines' similarity in this latent space as $k\left(\phi(p_\lambda^{(q)}; \theta), \phi(p_\lambda^{(\ell)}; \theta)\right)$ using the popular Matérn 5/2 kernel (Genton, 2002). Once we

compute the parameters of the kernel similarity function, we can compute the GP's posterior and conduct PO with BO as specified in Section 3.2.

In this work, we exploit existing deep kernel learning machinery (Wistuba & Grabocka, 2021; Wilson et al., 2016) to train the parameters $\theta$ of the pipeline embedding neural network $\phi$, and the parameters $\gamma$ of the kernel function $k$, by maximizing the log-likelihood of the observed validation losses $y$ of the evaluated pipeline configurations $p_\lambda$. The objective function for training a deep kernel is the log marginal likelihood of the Gaussian Process (Rasmussen & Williams, 2006) with covariance matrix entries $k_{q,\ell} = k\left(\phi(p_\lambda^{(q)};\theta), \phi(p_\lambda^{(\ell)};\theta)\right)$.

### 4.1 PIPELINE EMBEDDING NETWORK

The main piece of the puzzle is: How to define the pipeline configuration embedding $\phi$?

Our *DeepPipe* embedding is composed of two parts (i) per-algorithm encoders, and (ii) a pipeline aggregation network. A visualization example of our DeepPipe embedding architecture is provided in Figure 1. We define one encoder $\xi^{(i,j)}$ for the hyperparameter configurations of each $j$-th algorithm, in each $i$-th stage, as plain multi-layer perceptrons (MLP). These encoders, each parameterized by weights $\theta^{\text{enc}}_{(i,j)}$, map the algorithms' configurations to a $L_i$-dimensional vector space:

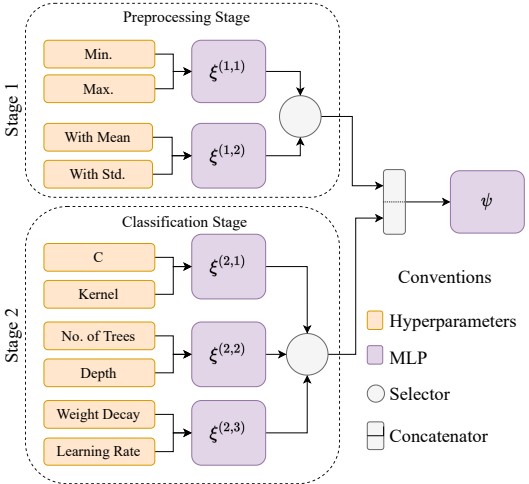

Figure 1: An example architecture for *DeepPipe* on a search space with 2 stages {Preprocessing, Classification}.

$$\xi^{(i,j)}\left(\lambda_{i,j}; \theta^{\text{enc}}_{i,j}\right) = \text{MLP}\left(\lambda_{i,j}; \theta^{\text{enc}}_{i,j}\right), \; \xi^{(i,j)}: \Lambda_{i,j} \to \mathbb{R}^{L_i}, \; \forall i \in \{1 \ldots N\}, \; \forall j \in \{1 \ldots M_i\} \quad (3)$$

For a pipeline configuration $p_\lambda$, represented with the indices of its algorithms $p$, and the configuration vectors of its algorithms $\lambda(p)$, we project all the pipeline's algorithms' configurations to their latent space using the algorithm-specific encoders. Then, we concatenate their latent encoder vectors, where our concatenation notation is $\mathbb{R}^{L_i} \oplus \mathbb{R}^{L_k} := \mathbb{R}^{L_i+L_k}$. Finally, the concatenated representation is embedded to a final $\mathbb{R}^Z$ space via an *aggregating* MLP $\psi$ with parameters $\theta^{\text{aggr}}$ as denoted below:

$$\phi\left(p_\lambda\right) := \psi\left(\xi^{(1,p_1)}(\lambda_{1,p_1}) \oplus \cdots \oplus \xi^{(N,p_N)}(\lambda_{N,p_N}) \mid \theta^{\text{aggr}}\right), \; \psi: \mathbb{R}^{\sum_i L_i} \to \mathbb{R}^Z \quad (4)$$

Within the $i$-th stage, only the output of one encoder is concatenated, therefore the output of the *Selector* corresponds to the active algorithm in the $i$-th stage and can be formalized as $\xi^{(i,p_i)}(\lambda_{i,p_i}) = \sum_{j=1}^{M_i} \mathbb{I}(j = a) \cdot \xi^{(i,j)}(\lambda_{i,j})$, where $a$ is the index of the active algorithm and $\mathbb{I}$ denotes the indicator function. Having defined the embedding $\phi$ in Equations 3-4, we can plug it into the kernel function, optimize it minimizing the negative log likelihood of the GP with respect to $\theta = \{\theta^{\text{enc}}, \theta^{\text{aggr}}\}$, and conduct BO as in Section 3.2.

### 4.2 META-LEARNING OUR PIPELINE EMBEDDING

In many practical applications, there exist computed evaluations of pipeline configurations on previous datasets, leading to the possibility of transfer learning for PO. Our *DeepPipe* can be easily meta-learned from such past evaluations by pre-training the pipeline embedding network. Let us denote the meta-dataset of pipeline evaluations on $T$ datasets (a.k.a. auxiliary tasks) as $\mathcal{H}_t = \{(p_\lambda^{(t,1)}, y^{(t,1)}), \ldots, (p_\lambda^{(t,Q_t)}, y^{(t,Q_t)})\}$, $t \in \{1, \ldots, T\}$, where $Q_t$ is the number of existing

evaluations for the $t$-th dataset. As a result, we meta-learn our method's parameters to minimize the meta-learning objective of Equation 5. This objective function corresponds to the negative marginal likelihood of the Gaussian Processes using *DeepPipe*'s extracted features as input to the kernel (Wistuba & Grabocka, 2021; Patacchiola et al., 2020). Further details on the meta-learning procedure of our pipeline configuration weights are provided in Appendix J.

$$\arg\min_{\gamma,\theta} \sum_{t=1}^{T} y^{(t)\mathrm{T}} K^{(t)}(\theta,\gamma)^{-1} y^{(t)} + \log \left| K^{(t)}(\theta,\gamma) \right| \tag{5}$$

## 5 EXPERIMENTS AND RESULTS

### 5.1 META-DATASETS

A meta-dataset is a collection of pipeline configurations and their respective performance evaluated in different tasks (i.e. datasets). Information about the meta-data sets is provided on Appendix P, their search spaces are clarified on Appendix R, and the splits of tasks per meta-dataset are found on Appendix T. All the tasks in the meta-dataset correspond to classification. We use the meta-training set for pre-training the Pipeline Optimization (PO) methods, the meta-validation set for tuning some of the hyper-parameters of the PO method, and we assess their performance on the meta-test set. In our experiments, we use the following meta-datasets.

**PMF** contains 38151 pipelines (after filtering out all pipelines with only NaN entries), and 553 datasets (Fusi et al., 2018). Although not all the pipelines were evaluated in all tasks, it has a total of 16M evaluations. The pipeline search space has 2 stages (preprocessing and estimator) with 2 and 11 algorithms respectively. Following the setup in the original paper (Fusi et al., 2018), we take 464 tasks for meta-training and 89 for meta-test. As the authors do not specify a validation meta-dataset, we sample randomly 15 tasks out of the meta-training dataset.

**Tensor-OBOE** provides 23424 pipelines and 551 tasks (Yang et al., 2020). It contains 11M evaluations, as there exist no evaluations for some combinations of pipelines and tasks. The pipelines include 5 stages: Imputator (1 algorithm), Dimensionality-Reducer (3 algorithms), Standardizer (1 algorithm), Encoder (1 algorithm), and Estimator (11 algorithms). We assign 331 tasks for meta-training tasks, 110 tasks for meta-validation, and 110 tasks for meta-testing.

**ZAP** is a benchmark that evaluates deep learning pipelines on fine-tuning state-of-the-art computer vision tasks (Ozturk et al., 2022). The meta-dataset contains 275625 evaluated pipeline configurations on 525 datasets and 525 different Deep Learning pipelines (i.e. the best pipeline of a dataset was evaluated also on all other datasets). From the set of datasets, we use 315 for meta-training, 45 for meta-validation and 105 for meta-test, following the protocol of the original paper.

In addition, we use **OpenML** datasets. It comprises 39 curated datasets (Gijsbers et al., 2019) and has been used in previous work for benchmarking (Erickson et al., 2020). This dataset does not contain pipeline evaluations like the other three meta-datasets above. However, we use the OpenML collection for evaluating the Pipeline Optimization in time-constrained settings (Ozturk et al., 2022).

### 5.2 BASELINES

**Random Search (RS)** selects pipeline configurations by sampling randomly from the search space (Bergstra & Bengio, 2012).

**Probabilistic Matrix Factorization (PMF)** uses a surrogate model that learns a latent representation for every pipeline using meta-training tasks (Fusi et al., 2018). We tuned this latent dimension for the Tensor-OBOE dataset from a grid of $\{10, 15, 20\}$ and found 20 to be the best setting. For the PMF-Dataset, where the model was introduced, we used the default value of 20. We use the original PMF implementation (Sheth, 2018).

**OBOE** also uses matrix factorization for optimizing pipelines, but they aim to find fast and informative algorithms to initialize the matrix (Yang et al., 2019). We use the settings provided by the authors.

**Tensor-OBOE** formulates PO as a tensor factorization, where the rank of the tensor is equal to $1 + N$, for $N$ being the number of stages in the pipeline (Yang et al., 2020). We tuned the rank for the Tucker decomposition from a grid of {20,30,40}, resulting in the best value being 30. All the other hyper-parameters were set as in the original implementation (Yang et al., 2019).

**Factorized Multilayer Perceptron (FMLP)** creates an ensemble of neural networks with a factorized layer (Schilling et al., 2015). The inputs of the neural network are the one-hot encodings of the algorithms and datasets, in addition to the algorithms' hyperparameters. We tuned the number of base estimators for the ensemble from a grid {10, 50, 100}, with 100 being the optimal ensemble size. Each network layer has 5 neurons and ReLU activations as highlighted in the author's paper (Schilling et al., 2015).

**RGPE** builds an ensemble of Gaussian Processes using auxiliary tasks (Feurer et al., 2018). The ensemble weights the contributions of every base model and the new model fit on the new task. We used the implementation from Botorch (Balandat et al., 2020).

**Gaussian Processes (GP)** are a standard and strong baseline in hyperparameter optimization (Snoek et al., 2012). We tuned the kernel from {Gaussian, Matérn 5/2}, with Matérn 5/2 performing better.

**DNGO** uses neural networks as basis functions with a Bayesian linear regressor at the output layer (Snoek et al., 2015). We use the implementation provided by Klein & Zela (2020), and its default hyperparameters.

**SMAC** uses Random Forest for predicting uncertainties (Hutter et al., 2011). After exploring a grid of {10, 50, 100} for the number of trees, we found 100 to be the best choice.

**TPOT** is an AutoML system that conducts PO using evolutionary search (Olson & Moore, 2016). We use the original implementation but adopted the search space to fit the Tensor-OBOE meta-dataset (see Appendix R).

### 5.3 RESEARCH HYPOTHESES AND ASSOCIATED EXPERIMENTS

**Hypothesis 1:** *DeepPipe* outperforms standard PO baselines.

**Experiment 1:** We evaluate the performance of *DeepPipe* when no meta-training data is available. We compare against four baselines: Random Search (RS) (Bergstra et al., 2011), Gaussian Processes (GP) (Rasmussen & Williams, 2006), DNGO (Snoek et al., 2015) and SMAC (Hutter et al., 2011). We evaluate their performances on the aforementioned PMF, Tensor-OBOE and ZAP meta-datasets. In Experiments 1 and 2 (below) we select 5 initial observations to warm-start the BO, then we run 95 additional iterations. The procedure for choosing these configurations is detailed in the Appendix G.

**Hypothesis 2:** Our meta-learned *DeepPipe* outperforms state-of-the-art transfer-learning PO methods.

**Experiment 2:** We compare our proposed method against baselines that use auxiliary tasks (a.k.a. meta-training data) for improving the performance of Pipeline Optimization: Probabilistic Matrix Factorization (PMF) (Fusi et al., 2018), Factorized Multilayer Perceptron (FMLP) (Schilling et al., 2015), OBOE (Yang et al., 2019) and Tensor OBOE (Yang et al., 2020). Moreover, we compare to RGPE (Feurer et al., 2018), an effective baseline for transfer HPO (Arango et al., 2021). We evaluate the performances on the PMF and Tensor-OBOE meta-datasets.

**Hypothesis 3:** *DeepPipe* leads to strong any-time results in a time-constrained PO problem.

**Experiment 3:** Oftentimes practitioners need AutoML systems that discover efficient pipelines within a small time budget. To test the convergence speed of our PO method we ran it on the aforementioned OpenML datasets for a budget of 10 minutes, and also 1 hour. We compare against five baselines: *(i)* TPOT (Olson & Moore, 2016) adapted to the search space of Tensor-OBOE (see Appendix R), *(ii)* OBOE and Tensor-OBOE (Yang et al., 2019; 2020) using the time-constrained version provided by the authors, *(iii)* SMAC (Hutter et al., 2011), and *(iv)* PMF (Fusi et al., 2018). The last three had the same five initial configurations used to warm-start BO as detailed in Experiment 1. Moreover, they were pre-trained with the Tensor-OBOE meta-dataset and all the method-specific settings are the same as in Experiment 2.

**Hypothesis 4:** Our novel encoder layers of *DeepPipe* enable an efficient PO when the pipeline search space changes, i.e. when developers add a new algorithm to a ML system.

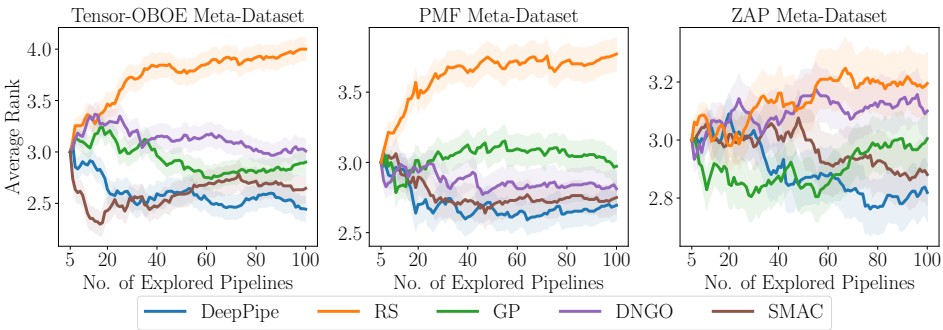

Figure 2: Comparison of *DeepPipe* vs. standard PO methods (Experiment 1)

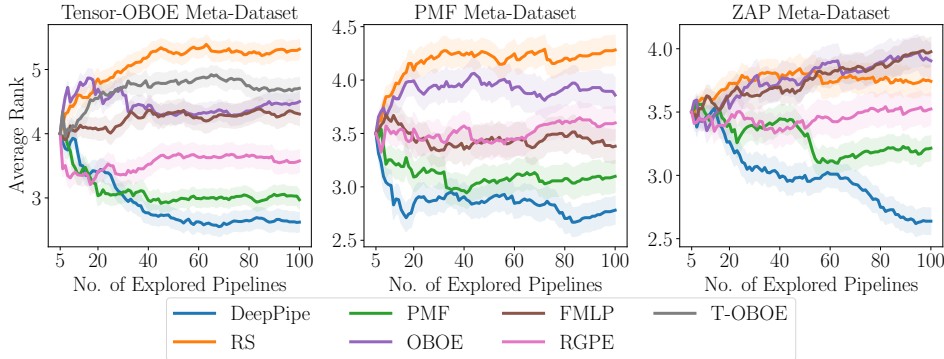

Figure 3: Comparison of *DeepPipe* vs. transfer-learning PO methods (Experiment 2)

**Experiment 4:** A major obstacle to meta-learning PO solutions is that they do not generalize when the search space changes, especially when the developers of ML systems add new algorithms. Our architecture quickly adapts to newly added algorithms **because only an encoder sub-network for the new algorithm should be trained**. To test the scenario, we ablate the performance of five versions of *DeepPipe* and try different settings when we remove a specific algorithm (an estimator) either from meta-training, meta-testing, or both.

**Hypothesis 5:** The encoders in *DeepPipe* introduce an inductive bias where latent representation vectors of an algorithm's configurations are co-located, and located distantly from the representations of other algorithms' configurations. Formally, given three pipelines $p^{(l)}, p^{(m)}, p^{(n)}$ if $p_i^{(l)} = p_i^{(m)}, p_i^{(l)} \neq p_i^{(n)}$ then $||\phi(p^{(l)}) - \phi(p^{(m)})|| < ||\phi(p^{(m)}) - \phi(p^{(n)})||$ with higher probability when using encoder layers, given that $p_i^{(n)}$ is the index of the algorithm in the $i$-th stage. Furthermore, we hypothesize that the less number of tasks during pre-training, the more necessary this inductive bias is.

**Experiment 5:** We sample 2000 pipelines of 5 estimation algorithms on the TensorOBOE dataset. Subsequently, we embed the pipelines using a *DeepPipe* with 0, 1 and 2 encoder layers, and weights $\theta$, initialized such that $\theta_i \in \theta$ are independently identically distributed $\theta_i \sim \mathcal{N}(0, 1)$. Finally, we visualize the embeddings with T-SNE (Van der Maaten & Hinton, 2008) and compute a cluster metric to assess how close pipelines with the same algorithm are in the latent space: $\mathbb{E}_{p^{(l)}, p^{(m)}, p^{(n)}} (\mathbb{I}(||\phi(p^{(l)}) - \phi(p^{(m)})|| < ||\phi(p^{(m)}) - \phi(p^{(n)})||))$. To test the importance of the inductive bias vs the number of pre-training tasks, we ablate the performance of *DeepPipe* for different percentage of pre-training tasks (0.5%, 1%, 5%, 10%, 50%, 100%) under different values of encoder layers.

## 5.4 EXPERIMENTAL SETUP FOR *DeepPipe*

The encoders and the aggregated layers are Multilayer Perceptrons with ReLU activations. We keep an architecture that is proportional to the input size. The number of neurons in the hidden layers for the encoder of algorithm $j$-th in $i$-th stage with $|\Lambda_{i,j}|$ hyperparameters is $F \cdot |\Lambda_{i,j}|$, given an integer factor $F$. The output dimension of the encoders of the $i$-th stage is defined as $Q_i = \max_j |\Lambda_{i,j}|$. The number of total layers (i.e. encoder and aggregation layers) is fixed to 4. The number of encoders is chosen from $\{0,1,2\}$. The specific values of the encoders' input dimensions are detailed in Appendix R. We choose $F \in \{4, 6, 8, 10\}$ based on the performance in the validation split. Specifically, we use the following values for *DeepPipe*: *(i)* in Experiment 1: 1 encoder layer (all meta-datasets), $F = 6$ (PMF and ZAP) and $F = 8$ (Tensor-OBOE), *(ii)* in Experiment 2: $F = 8$, no encoder layer (PMF, Tensor-OBOE) and one encoder layer (ZAP), *(iii)* in Experiment 3: $F = 8$ and no encoder layers, *(iv)* in Experiment 4 we use $F = 8$ and $\{0, 1\}$ encoder layers. Finally *(iv)* in Experiment 5 we use $F = 8$ and $\{0, 1, 2\}$ encoder layers. Additional details on the setup can be found in the Appendix G and our source code[2].

## 6 RESULTS

We present the results for Experiments 1 and 2 in Figures 2 and 3, respectively. In both cases, we compute the ranks of the classification accuracy achieved by the discovered pipelines of each technique, averaged across the meta-testing datasets. The shadowed lines correspond to the 95% confidence intervals. Additional results showing the mean regret are included in Appendix L. In Experiment 1 (standard/non-transfer PO) *DeepPipe* achieved the best performance for both meta-datasets, whereas SMAC attained the second place. In Experiment 2 *DeepPipe* strongly outperforms all the transfer-learning PO baselines in all meta-datasets. Given that *DeepPipe* yields state-of-the-art PO results on both standard and transfer-learning setups, we conclude that our pipeline embedding network computes efficient representations for PO with Bayesian Optimization. In particular, the results on the ZAP meta-dataset indicate the efficiency of *DeepPipe* in discovering **state-of-the-art Deep Learning pipelines for computer vision.**

Experiment 3 conducted on the OpenML datasets shows that *DeepPipe* performs well under restricted budgets (see Table 1). Although our method does not incorporate any direct way to handle time constraints, it is interesting to note that it outperforms other methods that include heuristics for handling a quick convergence, such as OBOE and Tensor-OBOE.

Table 1: Average Rank of Accuracy on OpenML Datasets

| Method | 10 Mins | 1 Hour |
|---|---|---|
| TPOT | $3.2000 \pm 0.1998$ | $3.3527 \pm 0.1911$ |
| Tensor-OBOE | $4.3878 \pm 0.1786$ | $4.3624 \pm 0.2008$ |
| OBOE | $3.9909 \pm 0.1935$ | $4.0873 \pm 0.2072$ |
| SMAC | $3.2424 \pm 0.1638$ | $3.1682 \pm 0.1484$ |
| PMF | $3.0424 \pm 0.1577$ | $2.9336 \pm 0.1523$ |
| *DeepPipe* | $\mathbf{2.7424 \pm 0.1277}$ | $\mathbf{2.8996 \pm 0.1343}$ |

Furthermore, the results of Experiment 4 indicate that our *DeepPipe* embedding quickly adapts to incrementally-expanding search spaces, e.g. when the developers of a ML system add new algorithms. In this circumstance, existing transfer-learning PO baselines do not adapt easily, because they assume a static pipeline search space. As a remedy, when a new algorithm is added to the system after meta-learning our pipeline embedding network, we train only a new encoder for that new algorithm. In this experiment, we run our method on variants of the search space when one algorithm at a time is introduced to the search space (for instance an estimator, e.g. MLP, RF, etc., is not known during meta-training, but added new to the meta-testing).

In Tables 2 and 5 (in Appendix), we present the values of the average rank among five different configurations for *DeepPipe*. We compare among meta-trained versions (denoted by ✓in the column *MTd.*) that omit specific estimators during meta-training (*MTr.*=✓), or during meta-testing (*MTe.*=✓). We also account for versions with one encoder layer denoted by ✓in the column *Enc*.

---

[2]The code is available in this repository: `https://anonymous.4open.science/r/DeepPipe-3DDF`

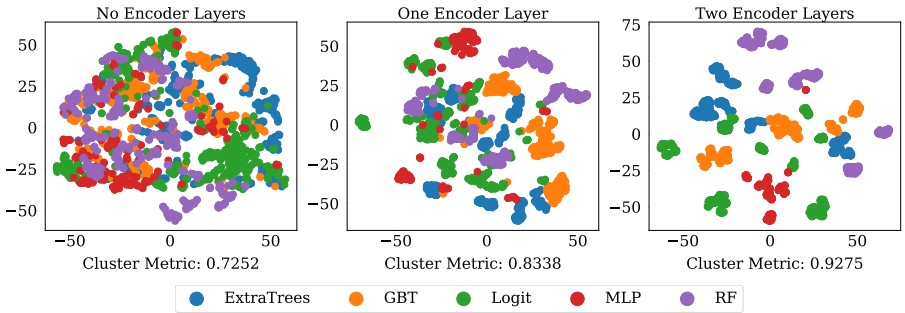

Figure 4: Embeddings of Pipelines produced by a random initialized *DeepPipe* (after applying T-SNE). The color indicates the active algorithm in the Estimation stage of Tensor-OBOE Meta-Dataset.

The best in all cases is the meta-learned model that did not omit the estimator (i.e. algorithm known and prior evaluations with that algorithm exist). Among the versions that omitted the estimator in the meta-training set (i.e. algorithm added new), the best configuration was the *DeepPipe* which fine-tuned a new encoder for that algorithm (line *Enc*=✓, *MTd.*=✓, *MTr.*=✓, *MTe.*=✗). This version of *DeepPipe* performs better than ablations with no encoder layers (i.e. only aggregation layers $\phi$), or the one omitting the algorithm during meta-testing (i.e. pipelines that do not use the new algorithm at all). The message of the results is simple: If we add a new algorithm to a ML system, instead of running PO without meta-learning (because the search space changes and existing transfer PO baselines are not applicable to the new space), we can use a meta-learned *DeepPipe* and only fine-tune an encoder for a new algorithm.

Table 2: Average rank among *DeepPipe* variants for newly-added algorithms (Tensor-OBOE)

| Enc. | MTd. | Omitted in | | Omitted Estimator | | | | | | | | | |
|---|---|---|---|---|---|---|---|---|---|---|---|---|---|
| | | MTr. | MTe. | ET | GBT | Logit | MLP | RF | lSVM | KNN | DT | AB | GB/PE |
| ✓ | ✓ | ✓ | ✓ | 3.2398 | 3.1572 | 3.0503 | 3.1982 | 3.4135 | 3.3589 | 3.2646 | 3.2863 | 3.1580 | 3.3117 |
| ✓ | ✗ | ✓ | ✗ | 3.5319 | 3.0934 | 3.6362 | 3.4780 | 3.4712 | 3.3829 | 3.6312 | 3.3691 | 3.6333 | 3.4642 |
| ✓ | ✓ | ✗ | ✗ | 2.5582 | 2.6773 | 2.7086 | 2.5761 | 2.6485 | 2.6938 | 2.6812 | 2.5596 | 2.5936 | 2.5546 |
| ✗ | ✓ | ✓ | ✗ | 2.9247 | 3.0743 | 2.8802 | 3.0423 | **2.6691** | 2.8026 | 2.7408 | 2.9161 | 2.9214 | 2.8689 |
| ✓ | ✓ | ✓ | ✗ | **2.7455** | **2.9978** | **2.7248** | **2.7054** | 2.7978 | **2.7619** | **2.6822** | **2.8688** | **2.6938** | **2.8007** |

The effect of the inductive bias introduced by the encoders (Experiment 5) can be appreciated in Figure 4. The pipelines with the same active algorithm in the estimation stage, but with different hyperparameters, lie closer in the embedding space created by a random initialized *DeepPipe*, forming compact clusters characterized by the defined cluster metric (value below the plots). We formally demonstrate in Appendix S that, in general, a single encoder layer is creating more compact clusters than a fully connected linear layer. In additional results (Appendix F), we observe that the average rank on the test-tasks improves for *DeepPipe* versions with deeper encoder layers (keeping the total number of layers fixed), if the number of meta-training tasks gets lower. This occurs because the objective function (Equation 5) makes possible to learn embeddings where pipelines with similar performance are clustered together (see Appendix O) given enough meta-training data. Otherwise, the inductive bias introduced by the encoders becomes more relevant.

## 7 CONCLUSION

In this paper, we have shown that efficient Machine Learning pipeline representations can be computed with deep modular networks. Such representations help discovering more accurate pipelines compared to the state-of-art approaches, because they capture the interactions of the different pipelines algorithms and their hyperparameters. Moreover, we show that introducing per-algorithm encoders helps in the case of limited meta-trained data, or when a new algorithm is added to the search space. Overall, we demonstrate that our method *DeepPipe* achieves the new state-of-the-art in Pipeline Optimization.

**Limitations.** Our representation network does not model complex parallel pipelines (in that case the embedding will be a graph neural network), and/or pipelines involving ensembles. We plan to investigate these important points in our future work.

**Reproducibility Statement**. To guarantee the reproducibility of or work, we include an anonymized repository to the related code. The code for the baselines is included also, and we reference the original implementations if it is the case. All the meta-datasets are publicly available and correspondingly referenced.

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

## A    POTENTIAL NEGATIVE SOCIETAL IMPACTS

The meta-training is the most demanding computational step, thus it can incur in high energy consumption. Additionally, *DeepPipe* does not handle fairness, so it may find pipelines that are biased by the data.

## B    LICENCE CLARIFICATION

The results of this work (code, data) are under license BSD-3-Clause license. Both PMF dataset Sheth (2018) and Tensor-OBOE dataset Akimoto & Yang (2020) hold the same license.

## C    DISCUSSION ON NUMBER OF EVALUATED PIPELINES

Based on results from Experiment 3, we report the average (and standard deviation) of the number of observed pipelines among all the compared methods in 10 and 60 minutes on Table 3. This is an important metric to understand the optimization overhead introduced by the method. For instance, a method that explores few pipelines during a fixed time window, might use expensive computations during the pipeline optimization.

Table 3: Average Number of Observed Pipelines on OpenML Datasets

| Method | 10 Mins | 1 Hour |
|---|---|---|
| TPOT | $45.48 \pm 46.25$ | $70.56 \pm 41.67$ |
| Tensor-OBOE | $84.43 \pm 57.61$ | $178.95 \pm 69.04$ |
| OBOE | $120.35 \pm 70.35$ | $467.09 \pm 330.34$ |
| SMAC | $80.76 \pm 115.04$ | $452.35 \pm 637.08$ |
| PMF | $126.37 \pm 197.61$ | $523.71 \pm 663.07$ |
| *DeepPipe* | $94.51 \pm 128.62$ | $356.71 \pm 379.62$ |

We notice that *DeepPipe* achieves the best results (see Table 1) by using a reasonable amount of pipelines, i.e. the optimization overhead introduced our method is small compared to other approaches such as TPOT and OBOE.

## D    DISCUSSION ON THE INTERACTIONS AMONG COMPONENTS

The encoder and aggregation layers capture interactions among the pipeline components, and therefore are important to attain good performance. These interactions are reflected in the features extracted by these layers, i.e. the pipelines representations obtained by *DeepPipe*. These representations lie on a metric space that captures relevant information about the pipelines and which can be used on the kernel for the Gaussian Process. Using the original input space does not allow to extract rich representations. To test this idea, we meta-train four version of *DeepPipe* with and without encoder and aggregation layers on our TensorOBOE meta-train set and then test on the meta-test split. In Figure 5, we show that the best version is obtained when using both encoder (*Enc.*) and aggregation (*Agg.*) layers (green line), whereas the worst version is obtained when using the original input space, i.e. no encoder and no aggregation layers. Having encoder helps more than not having en-

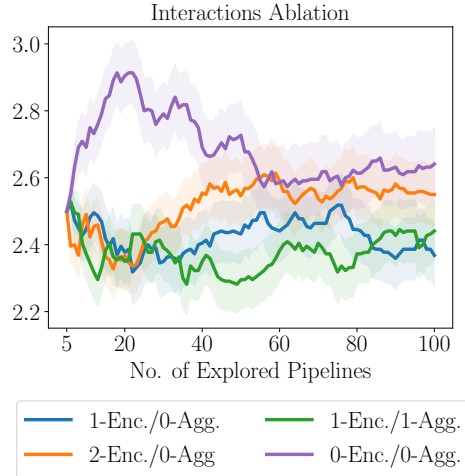

Figure 5: Average rank for *DeepPipe* with and without encoder and and aggregation layers.

coder, thus it is important to capture interactions among hyperparameters in the same stage. As having an aggregation layer is better than not, it is important to capture interactions among components from different stages.

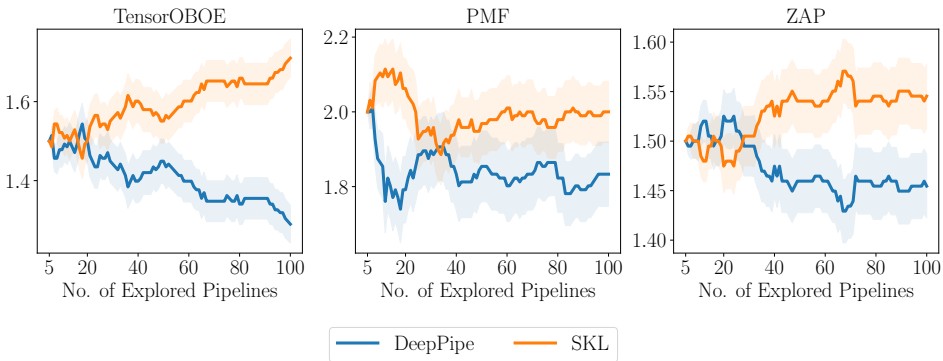

Figure 6: Comparison with Structured Kernel Learning (SKL)

Table 4: Comparison with AutoPrognosis

| $E_{BO}$ | Method | Avg. Rank | Std. Rank | Avg. Acc. | Std. Acc. | Avg Time | Std. Time |
|---|---|---|---|---|---|---|---|
| **50** | **AutoProg.** | 1.5588 | 0.4416 | 0.8637 | 0.1143 | 19324 | 12934 |
| | **DeepPipe** | 1.4411 | 0.4416 | 0.8692 | 0.1113 | 903 | 1548. |
| **100** | **AutoProg.** | 1.5133 | 0.4694 | 0.8715 | 0.0949 | 18502 | 11176 |
| | **DeepPipe** | 1.4866 | 0.4694 | 0.8727 | 0.0972 | 2221 | 5405 |

# E  COMPARISON WITH STRUCTURED KERNEL LEARNING (SKL) AND AUTOPROGNOSIS

AutoPrognosis (Alaa & van der Schaar, 2018) uses Structured Kernel Learning (SKL) and meta-learning to account for the interactions among the pipelines components. SKL decomposes the original input space by making up $N$ group of pipelines components, e.g. Random Forest and SVM in a group separated from Linear Regression and Logistic Regression. The hyperparameters of every group of pipelines components is then passed through a kernel, and then the $N$ resulting kernels are added. This effectively builds up a kernel with additive structure (Gardner et al., 2017), however they are not using a feature extractor like *DeepPipe*. We compare SKL against a non-pretrained *DeepPipe* on Figure 6 on three meta-datasets, where it is noticeable that our method outperforms this strategy.

Additionally we compare *DeepPipe* with the whole algorithm introduced by AutoPrognosis 2.0 (Imrie et al., 2022) on the Open ML datasets for 50 and 100 BO iterations ($E_{BO}$). We report the average and standard deviation for the rank, accuracy and time. *DeepPipe* achieves the best average rank, ie. lower average rank than AutoPrognosis. This is complemented with the having the highest average accuracy. Interestingly, our method is approximately one order of magnitude faster than AutoPrognosis. We notice this is due to the time overhead introduced by the Gibbs sampling strategy for optimizing the structured kernel, whereas our approach uses gradient-based optimization.

**Experimental Set-Up for *DeepPipe*.** For our comparison with SKL, we use the same hyperparameters and architecture as for the Experiment 1. When comparing with AutoPrognosis, we use the same hyperparmeters and architecture as for the Experiment 2, pre-trained on the Tensor-OBOE meta-train split.

**Experimental Set-Up for SKL and AutoPrognosis** For SKL we used the default strategy with $N = 3$ (Alaa & van der Schaar, 2018). For AutoPrognosis, we use the implementation in the respective author's repository [3]. We ran it with the default configuration, but limited the search space of classifiers to match the classifiers on the Tensor-OBOE meta-dataset [4].

---

[3]`https://github.com/ahmedmalaa/AutoPrognosis`

[4]Specifically, the list of classifiers is: Random Forest, Extra Tree Classifier, Gradient Boosting", Logist Regression, MLP, linear SVM, kNN, Decision Trees, Adaboost, Bernoulli Naive Bayes, Gaussian Naive Bayes, Perceptron.

## F    Discussion on the Inductive Bias vs. Pre-training Effect

How shallow/deep should the encoder networks be compared to the aggregation network? We hypothesize that deeper encoders help in the transfer-learning setup where there exist only a few evaluated pipeline configurations on past datasets. To test this hypothesis, we assess the performance of *DeepPipe* with different network sizes and meta-trained with different percentages of meta-training tasks: 0.5%, 1% , 5%, 10%, 50%, and 100%. As we use the Tensor-OBOE meta-dataset, this effectively means that we use 1, 3, 16, 33, 165, and 330 tasks respectively.

The results of these experiments are shown in Figure 7. Here we ablate *DeepPipe* with different numbers of encoder layers while pre-training on different fractions of the meta-training tasks. We ran the experiment for three values of $F$. The presented scores are the average ranks among the three *DeepPipe* configurations (row-wise). The average rank is computed across all the meta-test tasks and across 100 BO iterations. The results indicate that deeper encoders achieve a better performance when a small number of meta-training tasks is available. In contrast, shallower encoders are needed if more meta-training tasks are available. Apparently the deep aggregation layers $\phi$ already capture the in-

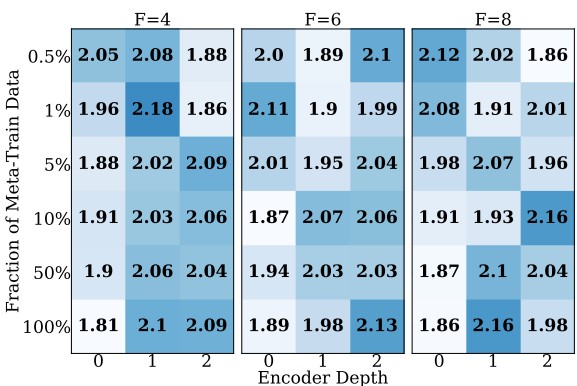

Figure 7: Comparison of the average rank for *DeepPipe* with different number of encoders under different percentages of meta-train data. The total number of layers is always the same.

teraction between the hyperparameter configurations across algorithms when a large meta-dataset of evaluated pipelines is given. The smaller the meta-data of evaluated pipeline configurations, the more inductive bias we need to implant in the form of per-algorithm encoders.

## G    Additional Information on Experimental Set-up

In all experiments (except Experiment 1), we meta-train the surrogate following Algorithm 1 in Appendix J for 10000 epochs with the Adam optimizer and a learning rate of $10^{-4}$, batch size 1000, and the Matérn Kernel. During meta-testing, when we perform BO to search for a pipeline, we fine-tune only the kernel parameters $\gamma$ for 100 gradient steps. In the non-transfer experiments (Experiment 1) we use an architecture with $F = 8$ and fine-tuned the network for 10000 iterations. The rest of the training settings are similar to the transfer experiments. In Experiment 5 we fine-tune the whole network for 100 steps when no encoders are used. Otherwise, we fine-tune only the encoder associated with the omitted estimator and freeze the rest of the network. We ran all experiments on a CPU cluster, where each node contains two Intel Xeon E5-2630v4 CPUs with 20 CPU cores each, running at 2.2 GHz. We reserved a total maximum memory of 16GB. We discuss how we implemented *DeepPipe* efficiently as a MLP with masked layers[5] in Appendix N. We associate algorithms with no hyperparameters to the same encoder. We found that adding the One-Hot-Encoding of the selected algorithms per stage as an additional input is helpful. Therefore, the input dimensionality of the aggregated layers is equal to the dimension after concatenating the encoders output $F \cdot \sum_i (Q_i + M_i)$. Further details on the architectures for each search space are specified in Appendix M. Finally, we use the Expected Improvement as acquisition function for *DeepPipe* and all the baselines.

**Initial Configurations**    For the experiments with the PMF-Dataset, we choose these configurations with the same procedure as the authors Fusi et al. (2018), where they use dataset meta-features to find the most similar auxiliary task to initialize the search on the test task. Since we do not have meta-features for the Tensor-OBOE meta-dataset, we follow a greedy initialization approach Metz et al. (2020). This was also applied to the ZAP-Dataset. Specifically, we select the best-performing

---

[5]We make our code available in `https://anonymous.4open.science/r/DeepPipe-E19E`

pipeline configuration by ranking their performances on the meta-training tasks. Subsequently, we iteratively choose four additional configurations that minimize $\sum_{t \in \text{Tasks}} \hat{r}_t$, where $\hat{r}_t = \min_{p \in \mathcal{X}} r_{t,p}$, given that $r_{t,p}$ is the rank of the pipeline $p$ on task $t$.

## H    ADDITIONAL RELATED WORK

**Hyperparameter Optimization (HPO)** has been well studied over the past decade (Bergstra & Bengio, 2012). Techniques relying on Bayesian Optimization (BO) employ surrogates to approximate the response function of Machine Learning models, such as Gaussian Processes (Snoek et al., 2012), Random Forests (Bergstra et al., 2011) or Bayesian Neural Networks (Snoek et al., 2015; Springenberg et al., 2016). Further improvements have been achieved by applying transfer learning, where existing evaluations on auxiliary tasks help pre-training or meta-learning the surrogate. In this sense, some approaches use pre-trained neural networks with uncertainty outputs (Wistuba & Grabocka, 2021; Perrone et al., 2018; Wei et al., 2021b), or ensembles of Gaussian Processes (Feurer et al., 2018).

**Deep Kernels** propose combining the benefits of stochastic processes such as Gaussian Processes with neural networks (Calandra et al., 2016; Garnelo et al., 2018; Wilson et al., 2016). Follow-up work has applied this combination for training few-shot classifiers (Patacchiola et al., 2020). In the area of Hyperparameter Optimization, (Snoek et al., 2015) achieved success on BO by modeling the output layer of a deep neural network with a Bayesian linear regression. Perrone et al. (2018) extended this work by pre-training the Bayesian network with auxiliary tasks. Recent work proposed to use non-linear kernels, such as the Matérn kernel, on top of the pre-trained network to improve the performance of BO (Wistuba & Grabocka, 2021; Wei et al., 2021a).

## I    BAYESIAN OPTIMIZATION (BO)

In BO we fit a surrogate iteratively using the observed configurations and their response. Posteriorly, its probabilistic output is used to query the next configuration to evaluate (observe) by maximizing an acquisition function. A common choice for the acquisition is Expected Improvement, defined as:

$$\text{EI}(p_\lambda | \mathcal{H}) = \mathbb{E}\left[\max\left\{\mu(p_\lambda) - y_{\max}, 0\right\}\right] \tag{6}$$

where $y_{\max}$ is the largest observed response in the history $\mathcal{H}$ and $\mu$ is the posterior of the mean predicted performance given by the surrogate, computed using Equation 2. A common choice as surrogate is Gaussian Process, but for Pipeline Optimization we introduce *DeepPipe*.

## J    *DeepPipe* META-TRAINING

Given a task $t$ with observations $\mathcal{H}_t = \{(p_\lambda^{(t,1)}, y^{(t,1)}), \ldots, (p_\lambda^{(t,Q_t)}, y^{(t,Q_t)})\}$, $t \in \{1, \ldots, T\}$, the objective function to minimize can be derived from the negative log marginal likelihood from the Gaussian Process $p(\mathcal{H}_t) \sim \mathcal{N}(0, K^T)$, where $K^{(t)}$ is the covariance matrix induced by *DeepPipe* with parameters $\theta, \gamma$. Specifically, the negative log marginal likelihood is (Rasmussen & Williams, 2006):

$$-\log p\left(\mathcal{H}_t\right) = -\log \mathcal{N}(0, K^{(t)}) = y^{(t)^{\text{T}}} K^{(t)}(\theta, \gamma)^{-1} y^{(t)} + \log\left|K^{(t)}(\theta, \gamma)\right| \tag{7}$$

The Equation 5 is the multi-task objective function that involves all the meta-learning tasks with indices $t \in \{1 \ldots, T\}$.

We use auxiliary tasks to learn a good initialization for the surrogate. We sample batches from the meta-training tasks, and make gradient steps that maximize the marginal log-likelihood in Equation 5, similar to previous work (Wistuba & Grabocka, 2021). The training algorithm for the surrogate is detailed in Algorithm 1. Additionally, we apply Early Convergence by monitoring the performance on the validation meta-dataset. Every epoch, we perform the following operations for every task $t \in 1 \ldots T$: *i)* Draw a set of $b$ observations (pipeline configuration and performance), *ii)* Compute the negative log marginal likelihood (our loss function) as in Equation 7, *iii)* compute gradient of the loss with respect to the *DeepPipe* parameters and *iv)* updated *DeepPipe* parameters.

---

**Algorithm 1:** *DeepPipe* Meta-Training

---

**Input:** Learning rates $\eta$, meta-training data with $T$ tasks $\mathcal{H} = \bigcup_{t=1..T} \mathcal{H}^{(t)}$, number of epochs $E$, batch size $b$
**Output:** Parameters $\mathbf{w}$ and $\boldsymbol{\theta}$
1 Initialize $\mathbf{w}$ and $\boldsymbol{\theta}$ at random;
2 **for** *E times* **do**
3     **for** $t \in \{1, ..., T\}$ **do**
4         Sample batch $\mathcal{B} = \{(p_\lambda^{(t,i)}, y^{(t,i)})\}_{i=1,...,b} \sim \mathcal{H}^{(t)}$;
5         Compute negative log-likelihood $\mathcal{L}$ on $\mathcal{B}$. (Objective Function in Equation 5);
6         $\theta^{\text{agg}} \leftarrow \theta^{\text{agg}} - \eta \nabla_{\theta^{\text{agg}}} \mathcal{L}$;
7         $\theta^{\text{enc}} \leftarrow \theta^{\text{enc}} - \eta \nabla_{\theta^{\text{enc}}} \mathcal{L}$;
8         $\gamma \leftarrow \gamma - \eta \nabla_{\gamma} \mathcal{L}$;
9     **end**
10 **end**

---

**Algorithm 2:** Bayesian Optimization (BO) with *DeepPipe*

---

**Input:** Learning rate $\eta$, initial observations $\mathcal{H} = \{(p_\lambda^{(i)}, y^{(i)})\}_{i=1,...,I}$, pretrained surrogate with parameters $\theta$ and $\gamma$, number of surrogate updates $E_{Test}$, BO iterations $E_{BO}$, search space of pipelines $\mathcal{P}$
**Output:** Pipeline Configuration $p_\lambda^*$
1 **Function** `FineTune` $(\mathcal{H}, \gamma, \eta, E_{test})$:
2     **for** $E_{Test}$ *times* **do**
3         Compute negative log-likelihood $\mathcal{L}$ on $\mathcal{D}$. (Objective function in Equation 5 with $T=1$);
4         $\gamma \leftarrow \gamma - \eta \nabla_{\gamma} \mathcal{L}$;
5     **end**
6     **return** $\gamma$
7 **Function** `BO` $(\mathcal{H}, \eta, \theta, \gamma, E_{test}, E_{BO})$:
8     **for** $E_{BO}$ *times* **do**
9         $\gamma' \leftarrow$ `FineTune`$(\mathcal{H}, \gamma, \eta, E_{Test})$;
10         Compute $p_\lambda' \in \arg\max_{p_\lambda \in \mathcal{P}} \text{EI}(p_\lambda, \gamma', \theta)$ ;
11         Observe performance $y'$ of pipeline $p_\lambda'$ ;
12         Add new observation $\mathcal{H} \leftarrow \mathcal{H} \cup \{(p_\lambda', y')\}$ ;
13     **end**
14     Compute best pipeline index $i_* \in \arg\max_{i \in \{1...|\mathcal{H}|\}} y_i$ ;
15     **return** $p_\lambda^{(i_*)}$ ;

---

## K   *DeepPipe* META-TESTING

When a new pipeline is to be optimized on a new dataset (task), we apply BO (see Algorithm 2). Every iteration we update the surrogate by fine-tuning the kernel parameters. However, the parameters of the MLP layers $\theta$ can be also optimized, as we did on the Experiment 1, in which case the parameters were randomly initialized.

## L   ADDITIONAL RESULTS

In this section, we present further results. Firstly, we show an ablation of the factor that determines the number of hidden units ($F$) in Figure 8. It shows that $F = 8$ attains the best performance after exploring 100 pipelines in both datasets. Additionally, we present the average regret for the ablation of $F$, and the results of Experiment 1 and 2 in Figures 9, 10 and 11 respectively.

Table 5 present the extended results of omitting estimators in the PMF Dataset. From these, we draw the same conclusion as in the same paper: having encoders help to obtain better performance when a new algorithm is added to a pipeline.

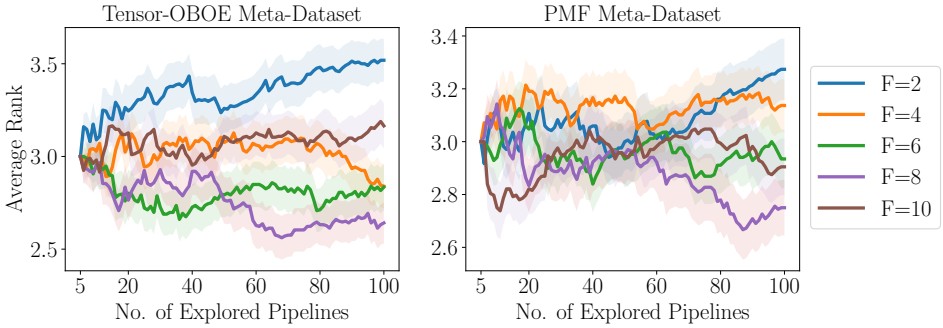

Figure 8: Comparison of different $F$ values in *DeepPipe* (Rank).

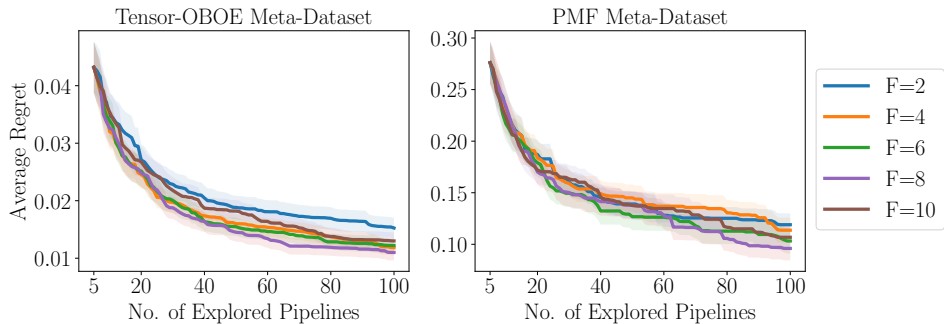

Figure 9: Comparison of different $F$ values in *DeepPipe* (Regret).

Table 5: Average rank among *DeepPipe* variants for newly-added algorithms (PMF)

| Enc. | MTd. | Omitted in | | Omitted Estimator | | | | | | | |
| | | MTr. | MTe. | ET | RF | XGBT | KNN | GB | DT | Q/LDA | NB |
| --- | --- | --- | --- | --- | --- | --- | --- | --- | --- | --- | --- |
| ✓ | ✓ | ✓ | ✓ | 3.1527 | 3.1645 | 3.2109 | 3.2541 | 3.2874 | 3.2741 | 3.1911 | 3.0263 |
| ✓ | ✗ | ✓ | ✗ | 3.2462 | 3.3208 | 3.2592 | 3.3180 | 3.2376 | 3.2249 | 3.3557 | 3.3993 |
| ✓ | ✓ | ✗ | ✗ | 2.5710 | 2.5996 | 2.4011 | 2.5947 | 2.6301 | 2.5664 | 2.6252 | 2.6214 |
| ✗ | ✓ | ✓ | ✗ | 3.0464 | **2.8550** | 3.0850 | **2.8845** | 2.9397 | 3.0316 | 2.9530 | 3.0596 |
| ✓ | ✓ | ✓ | ✗ | **2.9838** | 3.0601 | **3.0439** | 2.9486 | **2.9051** | **2.9029** | **2.8750** | **2.8934** |

We carry out an ablation to understand the difference between the versions of Deep Pipe with/without encoder and with/without transfer-learning using ZAP Meta-dataset. As shown in Figure 12, the version with transfer learning and one encoder performs the best, thus, highlighting the importance of encoders in transfer learning our *DeepPipe* surrogate.

## M ARCHITECTURE DETAILS

The input to the kernel has a dimensionality of $Z=20$. We fix it, to be the same as the output dimension for PMFs. The number of neurons per layer, as mentioned in the main paper, depends on $F$. Consider an architecture with with no encoder layers and $\ell_a$ aggregation layers, and hyperparameters $\Lambda_{i,j}, i \in \{1 \ldots N\}, j \in \{1 \ldots M_i\}$ (following the notation in section 4.1) with $Q_i = \max_j |\Lambda_{i,j}|$, then the number of weights (omitting biases for the sake of simplicity) will be:

$$\left( \sum_{i,j} |\Lambda_{i,j}| \right) \cdot \left( F \cdot \sum_i Q_i \right) + (\ell_a - 1) \left( F \cdot \sum_i Q_i \right)^2 \tag{8}$$

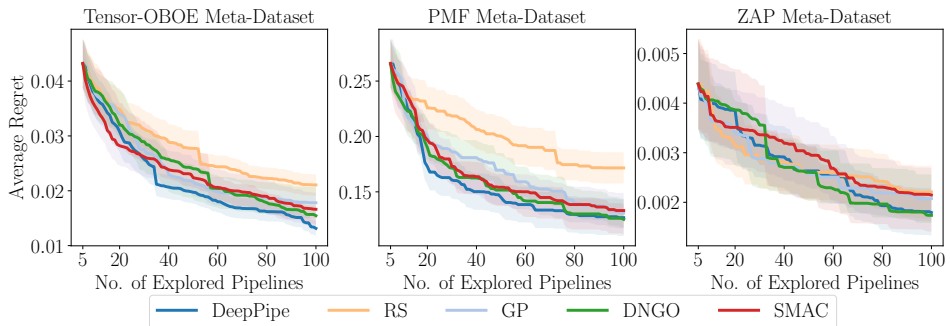

Figure 10: Comparison of *DeepPipe* vs. non transfer-learning PO methods in Experiment 1 (Regret)

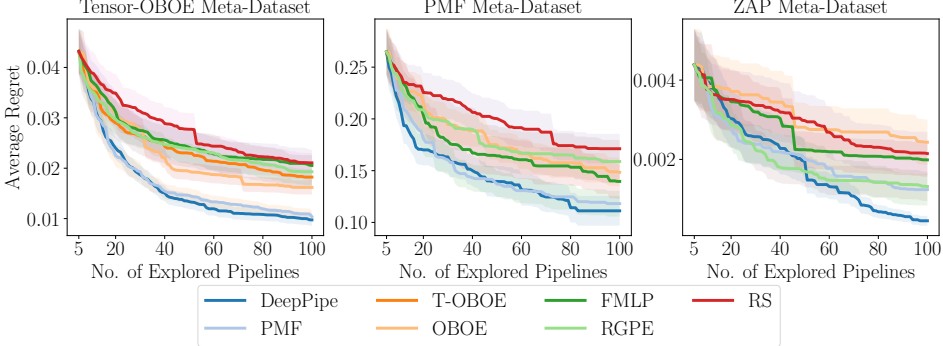

Figure 11: Comparison of Regret in *DeepPipe* vs. transfer-learning PO methods in Experiment 2 (Regret)

If the architecture has $\ell_e$ encoder layers and $\ell_a$ aggregation layers, then number of weights is given by:

$$\sum_{i,j} |\Lambda_{i,j}| \cdot (F \cdot Q_i) + (\ell_e - 1) \sum_i M_i \cdot (F \cdot Q_i)^2 + \ell_a \left( F \cdot \sum_i Q_i \right)^2 \tag{9}$$

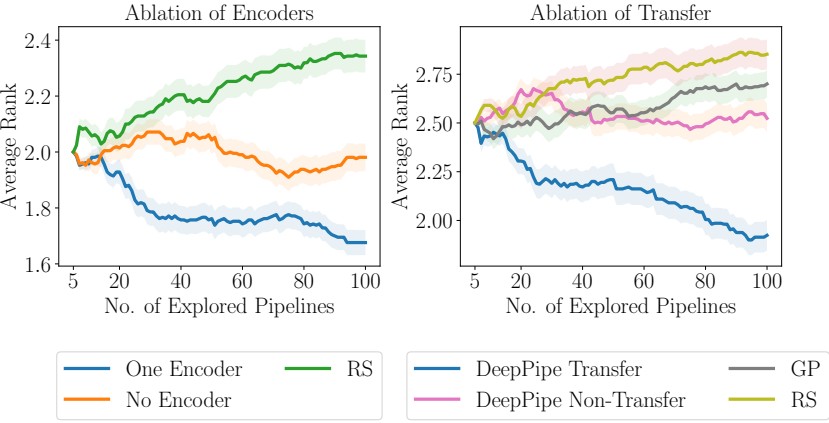

Figure 12: Ablations on the ZAP meta-dataset

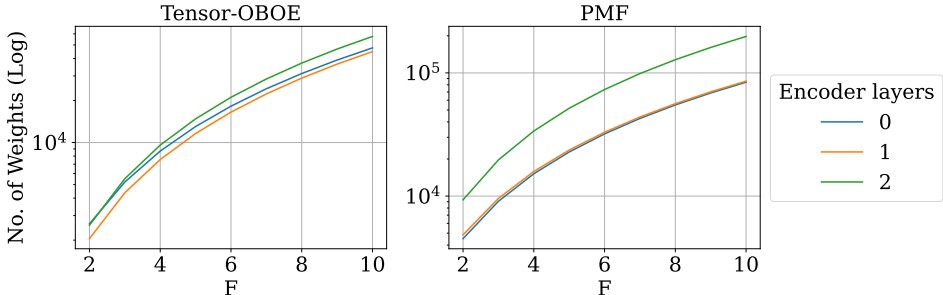

Figure 13: Number of weights in the MLP for a given value of $F$ and encoder layers.

In other words, the aggregation layers have $F \cdot \sum_i Q_i$ hidden neurons, whereas every encoder from the $i$-th stage has $F \cdot Q_i$ neurons per layer. The input sizes are $\sum_{i,j} |\Lambda_{i,j}|$ and $|\Lambda_{i,j}|$ for both cases respectively. The specific values for $|\Lambda_{i,j}|$ and $Q_i$ per search space are specified in Appendix R.

In the search space for PMF, we group the algorithms related to Naive Bayers (MultinomialNB, BernoulliNB, GaussianNB) in a single encoder. In this search space, we also group LDA and QDA. In the search space of TensorOboe, we group GaussianNB and Perceptron as they do not have hyperparameters. Given these considerations, we can compute the input size and the weights per search space as function of $\ell_a, \ell_e, F$ as follows:

(i) Input size:

$$
\begin{aligned}
\text{\# Input size (PMF)} &= \sum_{i,j} |\Lambda_{i,j}| = 72 \\
\text{\# Input (TensorOboe)} &= \sum_{i,j} |\Lambda_{i,j}| = 37 \\
\text{\# Input (ZAP)} &= \sum_{i,j} |\Lambda_{i,j}| = 35
\end{aligned}
\tag{10}
$$

(ii) Number of weights for architecture without encoder layers:

$$
\begin{aligned}
\text{\# Weights (PMF)} &= 720 \cdot F + 256 \cdot (\ell_a - 1) \cdot F^2 \\
\text{\# Weights (TensorOboe)} &= 444 \cdot F + 144 \cdot (\ell_a - 1) \cdot F^2 \\
\text{\# Weights (ZAP)} &= 1085 \cdot F + 961 \cdot (\ell_a - 1) \cdot F^2
\end{aligned}
\tag{11}
$$

(iii) Number of weights for architecture with encoder layers:

$$
\begin{aligned}
\text{\# Weights (PMF)} &= 886 \cdot F + (1376 \cdot (\ell_e - 1) + 256 \cdot \ell_a) \cdot F^2 \\
\text{\# Weights (TensorOboe)} &= 161 \cdot F + (271 \cdot (\ell_e - 1) + 144 \cdot \ell_a) \cdot F^2 \\
\text{\# Weights (ZAP)} &= 35 \cdot F + (965 \cdot (\ell_e - 1) + 961 \cdot \ell_a) \cdot F^2
\end{aligned}
\tag{12}
$$

According the previous formulations, Figure 13 shows how many parameters (only weights) the MLP has given a specific value of F and of encoder layers. We fix the total number of layers to four. Notice that the difference in the number of parameters between an architecture with 1 and 2 encoder layers is small in both search spaces.

## N  COMPUTATIONAL IMPLEMENTATION

*DeepPipe*'s architecture (encoder layers + aggregated layers) can be formulated as a Multilayer Perceptron (MLP) comprising three parts (Figure 14). The first part of the network that builds the

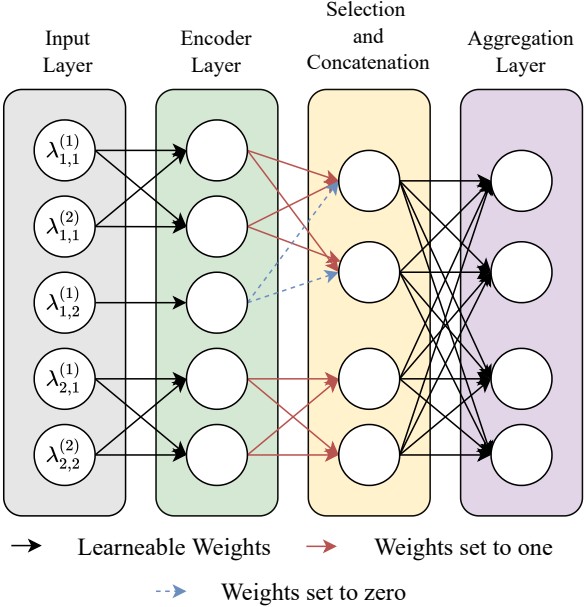

Figure 14: Example of the Implementation of *DeepPipe* as MLP. $\lambda_{i,j}^{(k)}$ indicates the $k$-th hyperparameter of the $j$-th algorithm in the $i$-th stage. In this architecture, the first stage has two algorithms, thus two encoders. The algorithm 1 is active for stage 1. The second stage has only one algorithm.

layers with encoders is implemented as a layer with masked weights. We connect the input values corresponding to the hyperparameters $\lambda_{(i,j)}$ of the $j$-th algorithm of the $i$-th stage to a fraction of the neurons in the following layer, what builds the encoder. The fraction of neurons, as explained in section 5.4 is $F \cdot \max_j |\lambda_{(i,j)}|$. The rest of the connections are dropped. The second part is a layer that selects the output of the encoders associated with the active algorithms (one per stage), and concatenates their outputs (*Selection & Concatenation*). The layer's connections are fixed to be either to one or zero during forward and backward pass. Specifically, they are one if they are connecting outputs of encoders of active algorithms, and zero otherwise. The last part, an *aggregation layer*, is a fully connected layer that learn interactions between the concatenated output of the encoders. By implementing the architecture as a MLP instead of a multiplexed list of nodes (e.g. with a module list in PyTorch), faster forward and backward passes are obtained. We only need to specify the selected algorithms in the forward-pass so that the weights in the *Encoder Layer* are masked and the ones in the *Selection & Concatenation* are accordingly set. After this implementation, notice that *DeepPipe* is a MLP with **sparse** connections.

## O    VISUALIZING THE LEARNT REPRESENTATIONS

We train a *DeepPipe* with 2-layer encoders, 2 aggregation layers, 20 output size and $F = 8$. To visualize the pipelines embeddings, we apply TSNE (T-distributed Stochastic Neighbor Embedding). As plotted in Figure 15, the pipelines with the same estimator and dimensionality reducer are creating clusters. The groups in this latent space are also indicators of the performance on a specific task. In Figure 16 we show the same embeddings of the pipelines with a color marker indicating its accuracy on two meta-testing tasks. Top-performing pipelines (yellow color) are relatively close to each other in both tasks, building up regions of good performing pipelines. These groups of good pipelines are different in both cases, which indicates that there is not a single pipeline that works for all tasks. *DeepPipe* maps the pipelines to an embedding space where it is easier to assess the similarity between pipelines and to search for good-performing pipelines. However, the type of pipeline (good performing pipelines, bad performing pipelines) depends on the task.

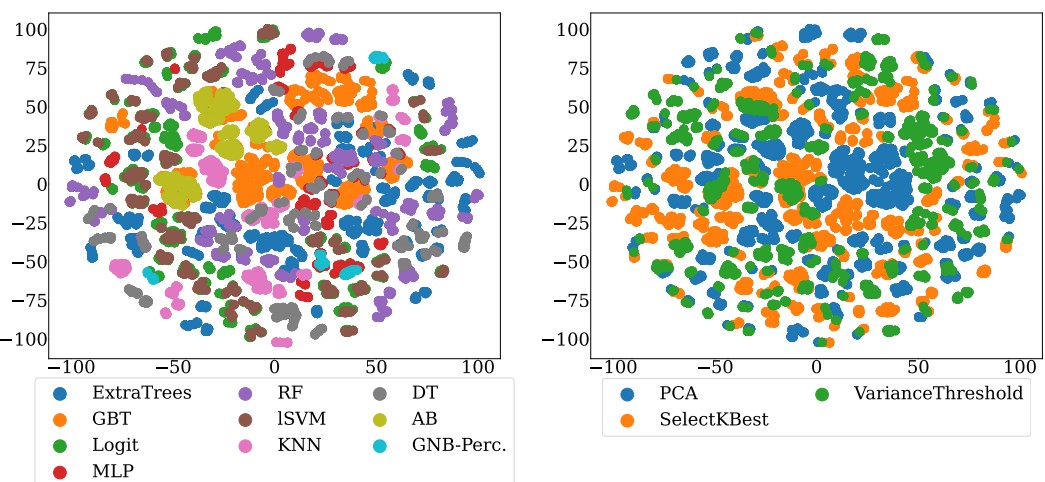

Figure 15: Learnt representations in 2 dimensions for estimators (left) and dimensionality reducers (right) from the Tensor-OBOE meta-dataset.

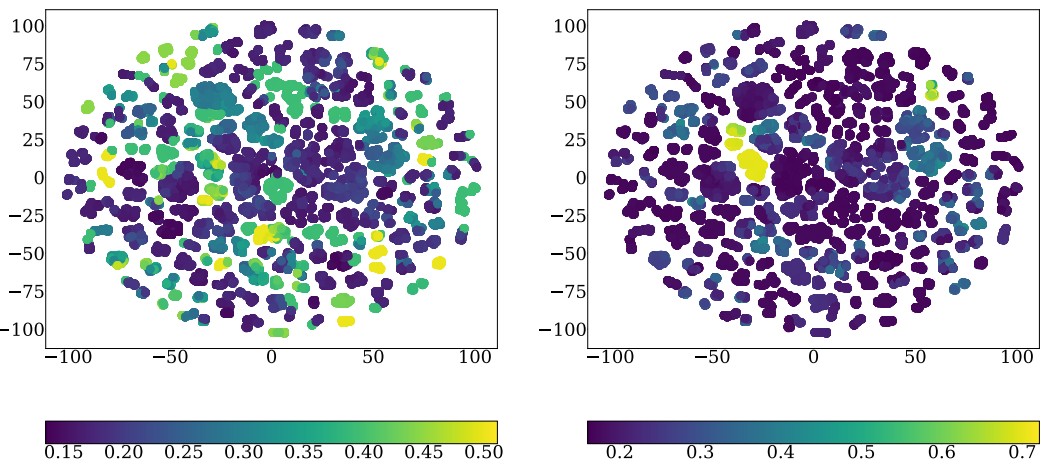

Figure 16: Learnt representations for two tasks with different accuracy levels.

## P    META-DATASET PREPROCESSING

We obtained the raw data for the meta-datasets from the raw repositories of PMF [6] and TensorOBOE [7]. PMF repo provides an accuracy matrix, while Tensor-OBOE specifies the error. We take the pipelines' configurations and concatenate the hyperparameters in both meta-datasets. Then we proceed with the following steps: 1) One-Hot encode the categorical hyperparameters, 2) apply a log transformation $x_{new} = \ln(x)$ to the hyperparameters whose value is greater than 3 standard deviations, 3) scale all the values to be in the range [0,1]. The variables coming from categorical hyperparameters are named *original-variable-name_category*.

## Q    ABBREVIATIONS

(i) Abbreviations in Table 2:

1) ET: ExtraTrees, 2) GBT: Gradient Boosting, 3) Logit: Logistict Regression 4) MLP: Multilayer Perceptron 5) RF: Random Forest, 6) lSVM: Linear Support Vector Machine, 7) kNN: k Nearest Neighbours, 8) DT: Decision Trees, 9) AB: AdaBoost, 10) GB/PE= Gaussian Naive Bayes/Perceptron.

(ii) Abbreviations in Table 3:

1) ET: ExtraTrees, 2) RF: Random Forest , 3) XGBT: Extreme Gradient Boosting, 4) kNN: K-Nearest Neighbours, 5) GB: Gradient Boosting, 6) DT: Decision Trees, 7) Q/LDA: Quadratic Discriminant Analysis/ Linear Discriminant Analysis, 8) NB: Naive Bayes.

## R    META-DATASET SEARCH SPACES

We detail the search spaces composition in Tables 6 and 7. We specify the stages, algorithms, hyperparameters, number of components per stage $M_i$, the number of hyperparameters per algorithm $|\lambda_{i,j}|$, and the maximum number of hyperparameters found in an algorithm per stage $Q_i$. For the ZAP meta-dataset, we defined a pipeline with two stages: (i) *Architecture*, which specifies the type or architecture used (i.e. ResNet18, EfficientNet-B0, EfficientNet-B1, EfficientNet-B2), and (ii) *Optimization-related Hyperparameters* that are shared by all the architectures.

---

[6]https://github.com/rsheth80/pmf-automl
[7]https://github.com/udellgroup/oboe/tree/master/oboe/defaults/TensorOboe

Table 6: Search Space for PMF Meta-Dataset

| Stage | $Q_i$ | $M_i$ | Algorithm | $|\Lambda_{i,j}|$ | Hyperparameters |
|---|---|---|---|---|---|
| Preprocessor | 3 | 2 | Polynomial | 3 | include_bias, interaction_only, degree |
| | | | PCA | 2 | keep_variance, whiten |
| Estimator | 13 | 8 | ExtraTrees | 9 | bootstrap, min_samples_leaf, n_estimators, max_features, min_weight_fraction_leaf, min_samples_split, max_depth |
| | | | RandomForest | 10 | bootstrap, min_samples_leaf, n_estimators, max_features, min_weight_fraction_leaf, min_samples_split, max_depth, criterion_entropy, criterion_gini |
| | | | XgradientBoosting | 13 | reg_alpha, col_sample_bytree, colsample_bylevel, scale_pos_weight, learning_rate, max_delta_step, base_score, n_estimators, subsample, reg_lambda, min_child_weight, max_depth, gamma |
| | | | kNN | 4 | p, n_neighbors, weights_distance, weights_uniform |
| | | | GradientBoosting | 10 | max_leaf_nodes, learning_rate, min_samples_leaf, n_estimators, subsample, min_weight_fraction_leaf, max_features, min_samples_split, max_depth, loss_deviance |
| | | | DecisionTree | 9 | max_leaf_nodes, min_samples_leaf, max_features, min_weight_fraction_leaf, min_samples_split, max_depth, splitter_best, criterion_entropy, criterion_gini |
| | | | LDA | 6 | shrinkage_factor, n_components, tol, shrinkage_-1, shrinkage_auto, shrinkage_manual |
| | | | QDA | 1 | reg_param |
| | | | BernoulliNB | 2 | alpha, fit_prior |
| | | | MultinomialNB | 2 | alpha, fit_prior |
| | | | GaussianNB | 1 | apply_gaussian_nb |

Table 7: Search Space for Tensor-OBOE Meta-Dataset

| Stage | $Q_i$ | $M_i$ | Algorithm | $|\Lambda_{i,j}|$ | Hyperparameters |
|---|---|---|---|---|---|
| Imputer | 4 | 1 | SimpleImputer | 4 | Strategy_constant, Strategy_mean, Strategy_median, Strategy_most_frequent |
| Encoder | 1 | 1 | OneHotEncoder | 1 | Handle_unknown_ignore |
| Scaler | 1 | 1 | StandardScaler | 1 | - |
| Dim. Reducer | 1 | 3 | PCA | 1 | N_components |
| | | | SelectKBest | 1 | K |
| | | | VarianceThreshold | 1 | - |
| Estimator | 5 | 10 | ExtraTrees | 3 | min_samples_split, criterion_entropy, criterion_gini |
| | | | Gradient Boosting | 4 | learning_rate, max_depth, max_features_None, max_features_log2 |
| | | | Logit | 5 | C, penalty_l1, penalty_l2, sovler_liblinear, solver_saga |
| | | | MLP | 5 | alpha, learning_rate_init, learning_rate_adaptive, solver_adam, solver_sgd |
| | | | Random Forest | 3 | min_samples_split, criterion_entropy, criterion_gini |
| | | | lSVM | 1 | C |
| | | | kNN | 2 | n_neighbors, p |
| | | | Decision Trees | 1 | min_samples_split |
| | | | AdaBoost | 2 | learning_rate, n_estimators |
| | | | GaussianNB | 1 | - |
| | | | Perceptron | 1 | - |

Table 8: Search Space for ZAP Meta-Dataset

| Stage | $Q_i$ | $M_i$ | Algorithm | $|\Lambda_{i,j}|$ | Hyperparameters |
|---|---|---|---|---|---|
| Architecture | 1 | 4 | ResNet | 1 | IsActive |
| | | | EfficientNet-B0 | 1 | IsActive |
| | | | EfficientNet-B1 | 1 | IsActive |
| | | | EfficientNet-B2 | 1 | IsActive |
| Common Hyperparameters | 31 | 1 | - | 31 | early_epoch, first_simple_model, max_inner_loop_ratio, skip_valid_score_threshold, test_after_at_least_seconds, test_after_at_least_seconds_max, test_after_at_least_seconds_step, batch_size, cv_valid_ratio, max_size, max_valid_count, steps_per_epoch, train_info_sample, optimizer.amsgrad, optimizer.freeze_portion, optimizer.lr, optimizer.min_lr, optimizer.momentum, optimizer.nesterov, optimizer.warm_up_epoch, warmup_multiplier, optimizer.wd, simple_model_LR, simple_model_NuSVC, simple_model_RF, simple_model_SVC, optimizer.scheduler_cosine, optimizer.scheduler_plateau, optimizer.type_Adam, optimizer.type_AdamW |

## S   THEORETICAL INSIGHT OF HYPOTHESIS 5

Here, we formally demonstrate that the *DeepPipe* with encoder layers is grouping hyperparameters from the same algorithm in the latent space, better than *DeepPipe* without encoders, formulated on Corollary S.4, which is supported by Proposition S.3.

**Lemma S.1.** *Given $\boldsymbol{w} \in \mathrm{R}^M$, a vector of weights with independent and identically distributed components $w_i \in \{w_1, ..., w_M\}$ such that $w_i \sim p(w)$, the expected value of the square of the norm $\mathbb{E}_{p(w)}(||\boldsymbol{w}||^2)$ is given by $M \cdot (\mu_w^2 + \sigma_w^2)$, where $\mu_w$ and $\sigma_w$ are the mean and standard deviation of $p(w)$ respectively.*

*Proof.*

$$\mathbb{E}_{p(w)}\left(||\boldsymbol{w}||^2\right) = \mathbb{E}_{p(w)}\left(\sum_{i=1}^{M} w_i^2\right) \tag{13}$$

$$= \sum_{i=1}^{M} \mathbb{E}_{p(w)}(w_i^2) \tag{14}$$

$$= \sum_{i=1}^{M} \mu_w^2 + \sigma_w^2 \tag{15}$$

$$= M \cdot (\mu_w^2 + \sigma_w^2) \tag{16}$$

$\square$

**Lemma S.2.** *Consider a linear function with scalar output $z = \boldsymbol{w}^T\boldsymbol{x}$ where $\boldsymbol{w} \in \mathrm{R}^{M \times 1}$ is the vector of weights with components $w_i, i \in \{1, ..., M\}$, $\boldsymbol{x} \in \mathrm{R}^{M \times 1}$ are the input features. Moreover, consider the weights are independently and identically distributed $w_i \sim p(w)$. The expected value of the norm of the output is given by $\mathbb{E}_{p(w)}\left(||\boldsymbol{w}^T\boldsymbol{x}||^2\right) = (\mu_w^2 + \sigma_w^2) \cdot ||\boldsymbol{x}||^2 + \mu_w^2 \cdot \sum_{i=1}^{M} \sum_{j=1}^{i-1} x_i \cdot x_j$.*

*Proof.*

$$\mathbb{E}_{p(w)}\left((\boldsymbol{w}^T\boldsymbol{x})^2\right) = \mathbb{E}_{p(w)}\left(\sum_{i=1}^{M} w_i \cdot x_i\right)^2 \tag{17}$$

$$= \mathbb{E}_{p(w)}\left(\sum_{i=1}^{M}(w_i \cdot x_i)^2 + \sum_{i=1}^{M}\sum_{j=1}^{i-1} w_i \cdot w_j \cdot x_i \cdot x_j\right) \tag{18}$$

$$= \sum_{i=1}^{M}\mathbb{E}_{p(w)}(w_i^2) \cdot x_i^2 + 2 \cdot \sum_{i=1}^{M}\sum_{j=1}^{i-1}\mathbb{E}_{p(w)}(w_i \cdot w_j) \cdot x_i \cdot x_j \tag{19}$$

$$\tag{20}$$

Since $w_i, w_j$ are independent then $\mathbb{E}_{p(w)}(w_i \cdot w_j) = \mathbb{E}_{p(w)}(w_i) \cdot \mathbb{E}_{p(w)}(w_j) = \mu_w^2$. Moreover, with a slight abuse in notation, we denote $\sum_{i=1}^{M}\sum_{j=1}^{i-1} x_i \cdot x_j = \boldsymbol{x} \otimes \boldsymbol{x}$. Given lemma S.1, we obtain:

$$\mathbb{E}_{p(w)}\left((\boldsymbol{w}^T\boldsymbol{x})^2\right) = (\mu_w^2 + \sigma_w^2) \cdot ||\boldsymbol{x}||^2 + 2 \cdot \mu_w^2 \cdot \boldsymbol{x} \otimes \boldsymbol{x} = D_w(\boldsymbol{x}) \tag{21}$$

$$\tag{22}$$

where $D_w(\cdot)$ is introduced as an operation to simplify the notation. $\square$

**Proposition S.3.** *Consider two vectors $\boldsymbol{x}', \hat{\boldsymbol{x}} \in \mathrm{R}^M$, and two weight vectors $\hat{\boldsymbol{w}}$ and $\boldsymbol{w}'$, $\hat{\boldsymbol{w}}^T\hat{\boldsymbol{x}} \in \mathrm{R}, {\boldsymbol{w}'}^T\boldsymbol{x}' \in \mathrm{R}$, such that the weights are iid. Then $\mathbb{E}_{p(w)}\left((\hat{\boldsymbol{w}}^T\hat{\boldsymbol{x}} - {\boldsymbol{w}'}^T\boldsymbol{x}')^2\right) > \mathbb{E}_{p(w)}\left((\hat{\boldsymbol{w}}^T\hat{\boldsymbol{x}} - \hat{\boldsymbol{w}}^T\boldsymbol{x}')^2\right)$.*

*Proof.* Using lemma S.2 and decomposition the argument within square:

$$\mathbb{E}_{p(w)}((\hat{\boldsymbol{w}}^T\hat{\boldsymbol{x}} - {\boldsymbol{w}'}^T\boldsymbol{x}')^2) = \mathbb{E}_{p(w)}\left((\hat{\boldsymbol{w}}^T\hat{\boldsymbol{x}})^2 + ({\boldsymbol{w}'}^T\boldsymbol{x}')^2 - 2 \cdot \hat{\boldsymbol{w}}^T\hat{\boldsymbol{x}} \cdot {\boldsymbol{w}'}^T\boldsymbol{x}'\right) \tag{23}$$

$$= D_w(\hat{\boldsymbol{x}}) + D_w(\boldsymbol{x}') - 2 \cdot \mathbb{E}_{p(w)}(\hat{\boldsymbol{w}}^T\hat{\boldsymbol{x}} \cdot {\boldsymbol{w}'}^T\boldsymbol{x}') \tag{24}$$

$$= D_w(\hat{\boldsymbol{x}}) + D_w(\boldsymbol{x}') - 2 \cdot \mathbb{E}_{p(w)}(\sum_{i=1}^{\hat{M}}\hat{w}_i \cdot \hat{x}_i \sum_{j=1}^{M'}w_j' \cdot x_j') \tag{25}$$

$$= D_w(\hat{\boldsymbol{x}}) + D_w(\boldsymbol{x}') - 2 \cdot \mathbb{E}_{p(w)}(\sum_{i=1}^{\hat{M}}\sum_{j=1}^{M'}w_j' \cdot x_j' \cdot \hat{w}_i \cdot \hat{x}_i) \tag{26}$$

$$= D_w(\hat{\boldsymbol{x}}) + D_w(\boldsymbol{x}') - 2 \cdot \sum_{i=1}^{\hat{M}}\sum_{j=1}^{M'}\mathbb{E}_{p(w)}(w_j' \cdot \hat{w}_i) \cdot x_j' \cdot \hat{x}_i \tag{27}$$

Since $\hat{\boldsymbol{w}}$ and $\boldsymbol{w}'$ are independent, then $\mathbb{E}_{p(w)}(w_j' \cdot \hat{w}_i) = \mathbb{E}_{p(w)}(w_j') \cdot \mathbb{E}_{p(w)}(\hat{w}_i) = \mu_w^2$. Thus,

$$\mathbb{E}_{p(w)}\left((\hat{\boldsymbol{w}}^T\hat{\boldsymbol{x}} - {\boldsymbol{w}'}^T\boldsymbol{x}')^2\right) = D_w(\hat{\boldsymbol{x}}) + D_w(\boldsymbol{x}') - 2 \cdot \mu_w^2 \cdot \sum_{i=1}^{\hat{M}}\sum_{j=1}^{M'}x_j' \cdot \hat{x}_i \tag{28}$$

When computing $\mathbb{E}_{p(w)}\left((\hat{\boldsymbol{w}}^T\hat{\boldsymbol{x}} - \hat{\boldsymbol{w}}^T\boldsymbol{x}')^2\right)$, we see that the weights are not independent, thus $\mathbb{E}_{p(w)}(\hat{w}_i \cdot \hat{w}_i) = \mu_w^2 + \sigma_w^2$, and

$$\mathbb{E}_{p(w)}\left((\hat{\boldsymbol{w}}^T\hat{\boldsymbol{x}} - \hat{\boldsymbol{w}}^T\boldsymbol{x}')^2\right) = D_w(\hat{\boldsymbol{x}}) + D_w(\boldsymbol{x}') - 2\cdot(\mu_w^2 + \sigma_w^2)\cdot\sum_{i=1}^{\hat{M}}\sum_{j=1}^{M'} x'_j\cdot\hat{x}_i \quad (29)$$

$$< D_w(\hat{\boldsymbol{x}}) + D_w(\boldsymbol{x}') - 2\cdot\mu_w^2\cdot\sum_{i=1}^{\hat{M}}\sum_{j=1}^{M'} {x_j}'\cdot\hat{x}_i \quad (30)$$

$$< \mathbb{E}_{p(w)}\left((\hat{\boldsymbol{w}}^T\hat{\boldsymbol{x}} - {\boldsymbol{w}'}^T\boldsymbol{x}')^2\right) \quad (31)$$

$\square$

**Corollary S.4.** *A random initialized DeepPipe with encoder layers induces an assumption that two hyperparameter configurations of an algorithm should have more similar performance than hyperparameter configurations from different algorithms.*

*Proof.* Given two hyperparameter configurations $\lambda^{(l)}, \lambda^{(m)}$ from an algorithm, and a third hyperparameter configuration $\lambda^{(n)}$ from a different algorithm, every random initialized encoder layer from *DeepPipe* maps the hyperparameters $\lambda^{(l)}, \lambda^{(m)}$ to latent dimensions $z^{(l)}, z^{(m)}$ that are closer to each other than to $z^{(n)}$, i.e. the expected distance among the output of the encoder layer will be $\mathbb{E}_{p(w)}(||z^l - z^m||) < \mathbb{E}_{p(w)}(||z^l - z^n||)$ based on Proposition S.3. Since *DeepPipe* uses a kernel such that $\kappa(\boldsymbol{x}, \boldsymbol{x}') = \kappa(\boldsymbol{x} - \boldsymbol{x}')$, their similarity will increase, when the distance between two configurations decreases. Thus, according to the Equation 2, they will have correlated performance. $\square$

## T    META-DATASET SPLITS

We specify the IDs of the task used per split. The ID of the tasks are taken from the original meta-dataset creators.

**(i) PMF Meta-Dataset**

**Meta-training:** 4538, 824, 1544, 1082, 1126, 917, 1153, 1063, 722, 1145, 1106, 1454, 4340, 477, 938, 806, 866, 333, 995, 1125, 924, 298, 755, 336, 820, 1471, 1120, 1520, 1569, 829, 958, 997, 472, 1442, 1122, 868, 313, 928, 921, 1446, 1536, 1025, 4534, 480, 723, 835, 1081, 950, 300, 1162, 821, 469, 933, 343, 766, 936, 1568, 785, 31, 164, 395, 761, 1534, 1056, 685, 1459, 230, 867, 828, 161, 742, 1136, 385, 877, 11, 1066, 1532, 1533, 941, 468, 1542, 795, 329, 792, 782, 1131, 796, 4153, 448, 1508, 1065, 1046, 1014, 54, 780, 748, 1150, 793, 1441, 1531, 717, 819, 1151, 287, 1016, 4135, 874, 162, 1148, 1005, 956, 1528, 23, 1516, 446, 1567, 41, 729, 910, 1156, 32, 1041, 1501, 955, 1129, 827, 937, 180, 1038, 973, 36, 44, 1496, 855, 400, 754, 1557, 1413, 758, 817, 1563, 181, 1127, 43, 444, 277, 1141, 715, 725, 884, 790, 880, 853, 155, 223, 1529, 1535, 6, 1009, 744, 1107, 1158, 830, 859, 947, 1475, 813, 734, 976, 227, 1137, 762, 777, 751, 784, 886, 885, 843, 1055, 1486, 1237, 225, 39, 778, 721, 392, 312, 857, 457, 1450, 209, 779, 479, 718, 801, 770, 1049, 391, 12, 730, 759, 1013, 338, 719, 988, 974, 787, 60, 741, 865, 1050, 735, 1079, 1482, 1143, 954, 1020, 1236, 814, 1048, 892, 879, 745, 971, 913, 1152, 694, 1133, 765, 905, 804, 848, 40477, 846, 334, 791, 923, 377, 1530, 889, 1163, 1006, 749, 922, 10, 59, 1541, 310, 461, 1538, 398, 870, 1481, 970, 1036, 1044, 1068, 187, 476, 1157, 40478, 1124, 1045, 845, 62, 915, 1167, 1059, 458, 815, 28, 797, 462, 21, 952, 467, 1505, 375, 882, 1011, 1460, 964, 1104, 275, 732, 189, 478, 1464, 979, 40474, 772, 720, 1022, 823, 811, 463, 61, 1451, 1067, 1165, 184, 716, 962, 978, 916, 1217, 935, 900, 925, 919, 871, 808, 335, 1457, 799, 983, 1169, 1004, 837, 1507, 4134, 890, 1062, 1510, 818, 728, 1135, 1147, 1019, 450, 1561, 40476, 816, 1562, 740, 864, 942, 151, 713, 953, 737, 1115, 1123, 1545, 1498, 850, 873, 959, 951, 987, 991, 1132, 1154, 294, 1040, 894, 26, 878, 307, 881, 746, 679, 872, 863, 943, 18, 1537, 767, 794, 1121, 1448, 401, 14, 1026, 833, 875, 1488, 383, 914, 20, 1043, 1116, 292, 847, 1540, 1069, 1155, 1015, 1238, 1149, 1546, 841, 1565, 1556, 1527, 682, 465, 1144, 769, 1517, 756, 834, 912, 807, 904, 16, 1061, 386, 805, 3, 775, 464, 50, 1455, 1021, 1160, 1140, 1489, 1519, 946, 994, 46, 22, 1443, 339, 969, 965, 30, 977, 860, 1500, 1064, 776, 822, 182, 743, 934, 1060, 803, 980, 1539, 346, 788, 1444, 1467, 727, 1509, 903, 832.

**Meta-Test:** 906, 789, 1159, 1600, 48, 1453, 876, 929, 1012, 891, 1164, 726, 459, 37, 812, 909, 927, 774, 278, 279, 1054, 918, 763, 394, 948, 40, 1100, 736, 1503, 1071, 1512, 1483, 53, 869, 285, 773,

1518, 197, 926, 836, 826, 907, 920, 1080, 1412, 276, 764, 945, 1543, 1472, 996, 908, 896, 851, 397, 783, 1084, 731, 888, 733, 1473, 753, 683, 893, 825, 902, 750, 1078, 8, 1073, 1077, 475, 724, 1513, 384, 388, 887, 714, 771, 1117, 1487, 337, 1447, 862, 838, 949, 800, 931, 911.

**Meta-Validation:** 1075, 747, 901, 1452, 389, 387, 752, 932, 768, 40475, 849, 1564, 1449, 895, 183.

**(ii) TensorOBOE Meta-Dataset**

**Meta-Training** 210, 20, 491, 339, 14, 170, 483, 284, 543, 220, 493, 64, 524, 485, 120, 81, 495, 362, 243, 545, 538, 532, 160, 541, 238, 436, 320, 272, 497, 412, 51, 195, 191, 116, 345, 400, 164, 106, 376, 63, 105, 308, 523, 490, 319, 93, 468, 517, 198, 145, 150, 39, 502, 364, 253, 303, 471, 2, 221, 518, 146, 241, 457, 114, 372, 176, 168, 536, 350, 338, 136, 416, 254, 337, 311, 464, 424, 255, 232, 133, 33, 88, 290, 44, 61, 199, 492, 529, 500, 343, 218, 302, 297, 73, 295, 35, 344, 29, 432, 410, 417, 309, 527, 217, 27, 402, 351, 156, 403, 414, 138, 212, 104, 438, 415, 421, 215, 466, 189, 214, 508, 204, 234, 259, 67, 24, 216, 300, 223, 129, 458, 111, 166, 505, 477, 40, 274, 427, 79, 375, 380, 327, 13, 287, 326, 496, 251, 228, 420, 161, 83, 117, 25, 110, 149, 152, 16, 407, 331, 109, 441, 422, 139, 237, 260, 352, 428, 317, 323, 484, 248, 449, 467, 19, 328, 296, 454, 269, 363, 226, 465, 3, 542, 125, 280, 286, 77, 184, 371, 455, 540, 275, 294, 521, 182, 32, 80, 307, 258, 11, 360, 447, 86, 266, 36, 193, 58, 41, 270, 411, 50, 209, 481, 480, 504, 503, 123, 222, 419, 62, 456, 377, 130, 187, 23, 451, 479, 43, 370, 394, 0, 383, 201, 405, 368, 515, 98, 387, 349, 304, 418, 292, 178, 369, 256, 94, 197, 95, 535, 163, 169, 69, 305, 48, 341, 373, 397, 207, 279, 514, 227, 148, 143, 334, 180, 356, 460, 131, 127, 47, 452, 262, 324, 203, 84, 426, 121, 544, 520, 534, 398, 384, 91, 82, 430, 267, 119, 358, 291, 57, 425, 487, 321, 257, 442, 42, 388, 335, 273, 488, 53, 522, 128, 28, 183, 459, 510, 151, 244, 265, 288, 423, 147, 177, 99, 448, 431, 115, 72, 537, 174, 87, 486, 314, 396, 472, 70, 277, 9, 359, 192

**Meta-Test** 118, 159, 548, 453, 385, 31, 512, 353, 247, 179, 332, 379, 10, 489, 112, 293, 219, 395, 281, 65, 409, 126, 401, 526, 342, 346, 413, 137, 366, 7, 381, 506, 289, 539, 282, 101, 97, 278, 54, 30, 298, 49, 100, 474, 461, 322, 283, 56, 144, 60, 6, 8, 507, 310, 336, 225, 261, 38, 329, 365, 445, 429, 513, 188, 469, 124, 154, 340, 59, 312, 473, 498, 546, 528, 263, 194, 55, 171, 236, 206, 158, 196, 34, 408, 18, 501, 250, 533, 52, 74, 26, 173, 92, 167, 4, 382, 181, 208, 354, 249, 450, 5, 141, 525, 200, 135, 531, 122, 22, 68

**Meta-Validation** 85, 446, 96, 172, 134, 37, 392, 90, 509, 389, 378, 435, 66, 391, 530, 333, 462, 231, 330, 301, 325, 268, 434, 318, 233, 213, 549, 140, 264, 482, 155, 235, 175, 157, 113, 165, 245, 246, 15, 361, 547, 470, 17, 306, 190, 153, 357, 45, 443, 162, 475, 186, 224, 494, 393, 399, 444, 550, 439, 516, 433, 230, 108, 89, 406, 46, 102, 463, 21, 107, 374, 211, 103, 71, 75, 316, 78, 240, 205, 386, 202, 142, 313, 252, 348, 511, 437, 347, 478, 355, 476, 242, 276, 519, 499, 285, 271, 229, 1, 390, 12, 132, 299, 404, 440, 239, 185, 76, 367, 315

**(iii) ZAP Meta-Dataset**

**Meta-Train** 0-svhn_cropped, 1-svhn_cropped, 2-svhn_cropped, 3-svhn_cropped, 4-svhn_cropped, 5-svhn_cropped, 6-svhn_cropped, 7-svhn_cropped, 8-svhn_cropped, 9-svhn_cropped, 10-svhn_cropped, 11-svhn_cropped, 12-svhn_cropped, 13-svhn_cropped, 14-svhn_cropped, 0-cycle_gan_apple2orange, 1-cycle_gan_apple2orange, 2-cycle_gan_apple2orange, 3-cycle_gan_apple2orange, 4-cycle_gan_apple2orange, 5-cycle_gan_apple2orange, 6-cycle_gan_apple2orange, 7-cycle_gan_apple2orange, 8-cycle_gan_apple2orange, 9-cycle_gan_apple2orange, 10-cycle_gan_apple2orange, 11-cycle_gan_apple2orange, 12-cycle_gan_apple2orange, 13-cycle_gan_apple2orange, 14-cycle_gan_apple2orange, 0-cats_vs_dogs, 1-cats_vs_dogs, 2-cats_vs_dogs, 3-cats_vs_dogs, 4-cats_vs_dogs, 5-cats_vs_dogs, 6-cats_vs_dogs, 7-cats_vs_dogs, 8-cats_vs_dogs, 9-cats_vs_dogs, 10-cats_vs_dogs, 11-cats_vs_dogs, 12-cats_vs_dogs, 13-cats_vs_dogs, 14-cats_vs_dogs, 0-stanford_dogs, 1-stanford_dogs, 2-stanford_dogs, 3-stanford_dogs, 4-stanford_dogs, 5-stanford_dogs, 6-stanford_dogs, 7-stanford_dogs, 8-stanford_dogs, 9-stanford_dogs, 10-stanford_dogs, 11-stanford_dogs, 12-stanford_dogs, 13-stanford_dogs, 14-stanford_dogs, 0-cifar100, 1-cifar100, 2-cifar100, 3-cifar100, 4-cifar100, 5-cifar100, 6-cifar100, 7-cifar100, 8-cifar100, 9-cifar100, 10-cifar100, 11-cifar100, 12-cifar100, 13-cifar100, 14-cifar100, 0-coil100, 1-coil100, 2-coil100, 3-coil100, 4-coil100, 5-coil100, 6-coil100, 7-coil100, 8-coil100, 9-coil100, 10-coil100, 11-coil100, 12-coil100, 13-coil100, 14-coil100, 0-omniglot, 1-omniglot, 2-omniglot, 3-omniglot, 4-omniglot, 5-omniglot, 6-omniglot, 7-omniglot, 8-omniglot, 9-omniglot, 10-omniglot, 11-omniglot, 12-omniglot, 13-omniglot, 14-omniglot, 0-cars196, 1-cars196, 2-cars196, 3-cars196, 4-cars196, 5-cars196, 6-cars196, 7-cars196, 8-cars196, 9-cars196, 10-cars196, 11-cars196, 12-cars196, 13-cars196,

14-cars196, 0-horses_or_humans, 1-horses_or_humans, 2-horses_or_humans, 3-horses_or_humans, 4-horses_or_humans, 5-horses_or_humans, 6-horses_or_humans, 7-horses_or_humans, 8-horses_or_humans, 9-horses_or_humans, 10-horses_or_humans, 11-horses_or_humans, 12-horses_or_humans, 13-horses_or_humans, 14-horses_or_humans, 0-tf_flowers, 1-tf_flowers, 2-tf_flowers, 3-tf_flowers, 4-tf_flowers, 5-tf_flowers, 6-tf_flowers, 7-tf_flowers, 8-tf_flowers, 9-tf_flowers, 10-tf_flowers, 11-tf_flowers, 12-tf_flowers, 13-tf_flowers, 14-tf_flowers, 0-cycle_gan_maps, 1-cycle_gan_maps, 2-cycle_gan_maps, 3-cycle_gan_maps, 4-cycle_gan_maps, 5-cycle_gan_maps, 6-cycle_gan_maps, 7-cycle_gan_maps, 8-cycle_gan_maps, 9-cycle_gan_maps, 10-cycle_gan_maps, 11-cycle_gan_maps, 12-cycle_gan_maps, 13-cycle_gan_maps, 14-cycle_gan_maps, 0-rock_paper_scissors, 1-rock_paper_scissors, 2-rock_paper_scissors, 3-rock_paper_scissors, 4-rock_paper_scissors, 5-rock_paper_scissors, 6-rock_paper_scissors, 7-rock_paper_scissors, 8-rock_paper_scissors, 9-rock_paper_scissors, 10-rock_paper_scissors, 11-rock_paper_scissors, 12-rock_paper_scissors, 13-rock_paper_scissors, 14-rock_paper_scissors, 0-cassava, 1-cassava, 2-cassava, 3-cassava, 4-cassava, 5-cassava, 6-cassava, 7-cassava, 8-cassava, 9-cassava, 10-cassava, 11-cassava, 12-cassava, 13-cassava, 14-cassava, 0-cmaterdb_devanagari, 1-cmaterdb_devanagari, 2-cmaterdb_devanagari, 3-cmaterdb_devanagari, 4-cmaterdb_devanagari, 5-cmaterdb_devanagari, 6-cmaterdb_devanagari, 7-cmaterdb_devanagari, 8-cmaterdb_devanagari, 9-cmaterdb_devanagari, 10-cmaterdb_devanagari, 11-cmaterdb_devanagari, 12-cmaterdb_devanagari, 13-cmaterdb_devanagari, 14-cmaterdb_devanagari, 0-cycle_gan_vangogh2photo, 1-cycle_gan_vangogh2photo, 2-cycle_gan_vangogh2photo, 3-cycle_gan_vangogh2photo, 4-cycle_gan_vangogh2photo, 5-cycle_gan_vangogh2photo, 6-cycle_gan_vangogh2photo, 7-cycle_gan_vangogh2photo, 8-cycle_gan_vangogh2photo, 9-cycle_gan_vangogh2photo, 10-cycle_gan_vangogh2photo, 11-cycle_gan_vangogh2photo, 12-cycle_gan_vangogh2photo, 13-cycle_gan_vangogh2photo, 14-cycle_gan_vangogh2photo, 0-cycle_gan_ukiyoe2photo, 1-cycle_gan_ukiyoe2photo, 2-cycle_gan_ukiyoe2photo, 3-cycle_gan_ukiyoe2photo, 4-cycle_gan_ukiyoe2photo, 5-cycle_gan_ukiyoe2photo, 6-cycle_gan_ukiyoe2photo, 7-cycle_gan_ukiyoe2photo, 8-cycle_gan_ukiyoe2photo, 9-cycle_gan_ukiyoe2photo, 10-cycle_gan_ukiyoe2photo, 11-cycle_gan_ukiyoe2photo, 12-cycle_gan_ukiyoe2photo, 13-cycle_gan_ukiyoe2photo, 14-cycle_gan_ukiyoe2photo, 0-cifar10, 1-cifar10, 2-cifar10, 3-cifar10, 4-cifar10, 5-cifar10, 6-cifar10, 7-cifar10, 8-cifar10, 9-cifar10, 10-cifar10, 11-cifar10, 12-cifar10, 13-cifar10, 14-cifar10, 0-cmaterdb_bangla, 1-cmaterdb_bangla, 2-cmaterdb_bangla, 3-cmaterdb_bangla, 4-cmaterdb_bangla, 5-cmaterdb_bangla, 6-cmaterdb_bangla, 7-cmaterdb_bangla, 8-cmaterdb_bangla, 9-cmaterdb_bangla, 10-cmaterdb_bangla, 11-cmaterdb_bangla, 12-cmaterdb_bangla, 13-cmaterdb_bangla, 14-cmaterdb_bangla, 0-cycle_gan_iphone2dslr_flower, 1-cycle_gan_iphone2dslr_flower, 2-cycle_gan_iphone2dslr_flower, 3-cycle_gan_iphone2dslr_flower, 4-cycle_gan_iphone2dslr_flower, 5-cycle_gan_iphone2dslr_flower, 6-cycle_gan_iphone2dslr_flower, 7-cycle_gan_iphone2dslr_flower, 8-cycle_gan_iphone2dslr_flower, 9-cycle_gan_iphone2dslr_flower, 10-cycle_gan_iphone2dslr_flower, 11-cycle_gan_iphone2dslr_flower, 12-cycle_gan_iphone2dslr_flower, 13-cycle_gan_iphone2dslr_flower, 14-cycle_gan_iphone2dslr_flower, 0-emnist_mnist, 1-emnist_mnist, 2-emnist_mnist, 3-emnist_mnist, 4-emnist_mnist, 5-emnist_mnist, 6-emnist_mnist, 7-emnist_mnist, 8-emnist_mnist, 9-emnist_mnist, 10-emnist_mnist, 11-emnist_mnist, 12-emnist_mnist, 13-emnist_mnist, 14-emnist_mnist, 0-eurosat_rgb, 1-eurosat_rgb, 2-eurosat_rgb, 3-eurosat_rgb, 4-eurosat_rgb, 5-eurosat_rgb, 6-eurosat_rgb, 7-eurosat_rgb, 8-eurosat_rgb, 9-eurosat_rgb, 10-eurosat_rgb, 11-eurosat_rgb, 12-eurosat_rgb, 13-eurosat_rgb, 14-eurosat_rgb, 0-colorectal_histology, 1-colorectal_histology, 2-colorectal_histology, 3-colorectal_histology, 4-colorectal_histology, 5-colorectal_histology, 6-colorectal_histology, 7-colorectal_histology, 8-colorectal_histology, 9-colorectal_histology, 10-colorectal_histology, 11-colorectal_histology, 12-colorectal_histology, 13-colorectal_histology, 14-colorectal_histology, 0-cmaterdb_telugu, 1-cmaterdb_telugu, 2-cmaterdb_telugu, 3-cmaterdb_telugu, 4-cmaterdb_telugu, 5-cmaterdb_telugu, 6-cmaterdb_telugu, 7-cmaterdb_telugu, 8-cmaterdb_telugu, 9-cmaterdb_telugu, 10-cmaterdb_telugu, 11-cmaterdb_telugu, 12-cmaterdb_telugu, 13-cmaterdb_telugu, 14-cmaterdb_telugu, 0-uc_merced, 1-uc_merced, 2-uc_merced, 3-uc_merced, 4-uc_merced, 5-uc_merced, 6-uc_merced, 7-uc_merced, 8-uc_merced, 9-uc_merced, 10-uc_merced, 11-uc_merced, 12-uc_merced, 13-uc_merced, 14-uc_merced, 0-kmnist, 1-kmnist, 2-kmnist, 3-kmnist, 4-kmnist, 5-kmnist, 6-kmnist, 7-kmnist, 8-kmnist, 9-kmnist, 10-kmnist, 11-kmnist, 12-kmnist, 13-kmnist, 14-kmnist

**Meta-Test** 0-cycle_gan_summer2winter_yosemite, 1-cycle_gan_summer2winter_yosemite, 2-cycle_gan_summer2winter_yosemite, 3-cycle_gan_summer2winter_yosemite, 4-cycle_gan_summer2winter_yosemite, 5-cycle_gan_summer2winter_yosemite, 6-

cycle_gan_summer2winter_yosemite, 7-cycle_gan_summer2winter_yosemite, 8-cycle_gan_summer2winter_yosemite, 9-cycle_gan_summer2winter_yosemite, 10-cycle_gan_summer2winter_yosemite, 11-cycle_gan_summer2winter_yosemite, 12-cycle_gan_summer2winter_yosemite, 13-cycle_gan_summer2winter_yosemite, 14-cycle_gan_summer2winter_yosemite, 0-malaria, 1-malaria, 2-malaria, 3-malaria, 4-malaria, 5-malaria, 6-malaria, 7-malaria, 8-malaria, 9-malaria, 10-malaria, 11-malaria, 12-malaria, 13-malaria, 14-malaria, 0-cycle_gan_facades, 1-cycle_gan_facades, 2-cycle_gan_facades, 3-cycle_gan_facades, 4-cycle_gan_facades, 5-cycle_gan_facades, 6-cycle_gan_facades, 7-cycle_gan_facades, 8-cycle_gan_facades, 9-cycle_gan_facades, 10-cycle_gan_facades, 11-cycle_gan_facades, 12-cycle_gan_facades, 13-cycle_gan_facades, 14-cycle_gan_facades, 0-emnist_balanced, 1-emnist_balanced, 2-emnist_balanced, 3-emnist_balanced, 4-emnist_balanced, 5-emnist_balanced, 6-emnist_balanced, 7-emnist_balanced, 8-emnist_balanced, 9-emnist_balanced, 10-emnist_balanced, 11-emnist_balanced, 12-emnist_balanced, 13-emnist_balanced, 14-emnist_balanced, 0-imagenette, 1-imagenette, 2-imagenette, 3-imagenette, 4-imagenette, 5-imagenette, 6-imagenette, 7-imagenette, 8-imagenette, 9-imagenette, 10-imagenette, 11-imagenette, 12-imagenette, 13-imagenette, 14-imagenette, 0-mnist, 1-mnist, 2-mnist, 3-mnist, 4-mnist, 5-mnist, 6-mnist, 7-mnist, 8-mnist, 9-mnist, 10-mnist, 11-mnist, 12-mnist, 13-mnist, 14-mnist, 0-cycle_gan_horse2zebra, 1-cycle_gan_horse2zebra, 2-cycle_gan_horse2zebra, 3-cycle_gan_horse2zebra, 4-cycle_gan_horse2zebra, 5-cycle_gan_horse2zebra, 6-cycle_gan_horse2zebra, 7-cycle_gan_horse2zebra, 8-cycle_gan_horse2zebra, 9-cycle_gan_horse2zebra, 10-cycle_gan_horse2zebra, 11-cycle_gan_horse2zebra, 12-cycle_gan_horse2zebra, 13-cycle_gan_horse2zebra, 14-cycle_gan_horse2zebra

**Meta-Validation** 0-emnist_byclass, 1-emnist_byclass, 2-emnist_byclass, 3-emnist_byclass, 4-emnist_byclass, 5-emnist_byclass, 6-emnist_byclass, 7-emnist_byclass, 8-emnist_byclass, 9-emnist_byclass, 10-emnist_byclass, 11-emnist_byclass, 12-emnist_byclass, 13-emnist_byclass, 14-emnist_byclass, 0-imagenet_resized_32x32, 1-imagenet_resized_32x32, 2-imagenet_resized_32x32, 3-imagenet_resized_32x32, 4-imagenet_resized_32x32, 5-imagenet_resized_32x32, 6-imagenet_resized_32x32, 7-imagenet_resized_32x32, 8-imagenet_resized_32x32, 9-imagenet_resized_32x32, 10-imagenet_resized_32x32, 11-imagenet_resized_32x32, 12-imagenet_resized_32x32, 13-imagenet_resized_32x32, 14-imagenet_resized_32x32, 0-fashion_mnist, 1-fashion_mnist, 2-fashion_mnist, 3-fashion_mnist, 4-fashion_mnist, 5-fashion_mnist, 6-fashion_mnist, 7-fashion_mnist, 8-fashion_mnist, 9-fashion_mnist, 10-fashion_mnist, 11-fashion_mnist, 12-fashion_mnist, 13-fashion_mnist, 14-fashion_mnist

## (iv) OpenML Datasets

10101, 12, 146195, 146212, 146606, 146818, 146821, 146822, 146825, 14965, 167119, 167120, 168329, 168330, 168331, 168332, 168335, 168337, 168338, 168868, 168908, 168909, 168910, 168911, 168912, 189354, 189355, 189356, 3, 31, 34539, 3917, 3945, 53, 7592, 7593, 9952, 9977, 9981

We checked that there is not overlap between the tasks used for meta-training from the TensorOBOE and the tasks used on OpenML Datasets.

