# OpenReview forum: "DeepPipe: Deep, Modular and Extendable Representations of Machine Learning Pipelines"
_ICLR.cc/2023/Conference — Submitted to ICLR 2023_

### Official Review · Reviewer_5R6t · 2022-10-23

**Confidence:** 4
**Clarity, Quality, Novelty And Reproducibility:** 1. The novelty of this paper is limit…
**Correctness:** 3
**Technical Novelty And Significance:** 2
**Empirical Novelty And Significance:** 3
**Recommendation:** 6

**Strength And Weaknesses:**

Strengths:
1. The paper is nicely structured and clearly written.
2. The experiments carried out cover rather extensive scenarios.
3. The authors additionally looked into the inductive bias of the introduced embeddings, which results in some insights for the architecture design.

Weaknesses:
1. The novelty of the method is limited, in the context of recent literature.
2. Lack of empirical justification for the proposed method in the general deep learning setting, where pipeline optimization can be of great importance.

**Summary Of The Paper:**

This paper aims to tackle the pipeline optimization problem that jointly optimizes various hyperparameters in the whole pipeline of machine learning. It proposes per-algorithm encoders and further aggregates the embeddings from all stages of the pipeline to a vector representation. Bayesian optimization is able to efficiently optimize the pipeline performance using the aggregated representation. The paper empirically shows the proposed method’s better performance against baselines. The authors also show the possibility of meta-training the model on existing pipeline evaluation data, and the method’s ability to quickly adapt to different dataset tasks as well as changes in the pipeline structure (e.g., the addition of new algorithms).

**Summary Of The Review:**

Overall, the paper conducts many empirical experiments to validate the superior performance of DeepPipe. However, several aspects are lacking. The authors did not fully establish the gap between the literature and this paper. The inability of previous methods in capturing the interaction and relations among pipeline stages is hypothesized, rather than proved or shown in the experiments. Some important baselines are left out in the experiments.

---

> ### Author Response · Authors · 2022-11-16
> **Reply to Reviewer 5R6t**
>
> Dear Reviewer,
>
> Thank you for your observations and remarks. We address each of them by referring to the new updated version of the paper:
>
> > 1. **Reviewer's Question:** *The novelty of this paper is limited as 1) meta-learning approaches for GP deep kernels have been investigated in previous works (Wistuba & Grabocka, 2021) and 2) kernel learning approaches for pipeline learning have been studied as well (Alaa & van der Schaar, 2018). The only difference could be that this work uses an additional fully-connected layer on top of the component-wise embeddings. However, why is this essential? Can BO already consider and utilize the similarities and interactions between these embeddings?*
>
> 1. **Author's Answer:** Thank you for this interesting remark. Previous work  (Wistuba & Grabocka, 2021) did not study deep kernels in the context of Pipeline optimization, and they did not study the effect of sets of encoders and aggregation layers. Moreover, they did not study how to extend the deep kernel for new algorithms with new hyperparameters. The other suggested kernel approach for pipeline learning (Alaa & van der Schaar, 2018) does not use deep kernels, and they rely on finding shallow kernels (on raw hyperparameter spaces) with additive structures. Our work shows that we need several layers for capturing the similarity of pipelines by learning a deep kernel with an inductive bias through the architecture design (exploiting the knowledge of the stage-wise pipeline construction), and we need meta-learning for capturing these interactions from evaluations on previous datasets. Please see our new appendix Appendix D discussing the interactions among components through a new extensive ablation study.
>
>
> > 2. **Reviewer's Question:** *Related to the point above, the highly relevant papers mentioned are not used as the baseline for comparison in the experiment section. The authors should at least compare to (Alaa & van der Schaar, 2018) to investigate the effectiveness of pipeline embedding networks, which is the main part that differentiates from other papers.*
>
> 2. **Author's Answer:** You have a very valid point. During the rebuttal period, we conducted experiments to compare DeepPipe vs Structured Kernel Learning (the main method used by AutoPrognosis), and the AutoPrognosis package provided by the authors.  We also measure the runtime time (overall time for pipeline optimization) in this experiment. We found that DeepPipe performs better than AutoPrognosis and runs 100 BO iterations 10 times faster than AutoPrognosis. Their Bayesian sampling approach for finding the kernel structure is very time-consuming compared to our method. Please find the detailed results and the associated discussion in Appendix E of the new manuscript or link: https://anonymous.4open.science/r/DeepPipe-3DDF/Comparison_with_AutoPrognosis.md
>
> > 3. **Reviewer's Question:** *Does the budget in Experiment 3 include the model training time, or is it just the search cost? Can you accompany it with the number of evaluations done to get a sense of the search time overhead, especially when DeepPipe needs to perform fine-tuning?*
>
> 3. **Author's Answer:** We added in Appendix C measurements on the number of pipelines explored in the time-constrained experiments. The results show that DeepPipe does not have a noticeably larger computational overhead compared to other methods, and evaluates a comparable number of evaluations within the same time budget.
>
> > 4. **Reviewer's Question:** *For the scenario when new algorithms are added in Experiment 4, does the aggregator also needs to be retrained? Also, have you considered cases where new stages are added?*
>
> 4. **Author's Answer:** We just need to train the new encoder, and freeze the aggregator layer.
>
> > 5. **Reviewer's Question:** *T-OBOE’s results missing in Experiment 2. Figure 3 and Figure 9.*
>
> 5. **Author's Answer:**  We did not include T-OBOE for the PMF and ZAP meta-dataset, because the performance matrices of these datasets do not factorize into a tensor that fits memory. Tensor OBOE uses a tensor $T \in \mathrm{R}^{E_1 \times E_2 \times M}$, where $E_1, E_2$ is the number of unique configurations for stage 1 and 2, from M meta-training tasks. The authors consider configuration from a continuous range, instead of a discrete grid (like in the TensorOBOE meta-dataset), thus the number of unique values per HP is large, and the number of unique configurations per stage is also very high.
> We exemplify this with real numbers from PMF. The preprocessing stage has 36950 unique configurations, while the estimation stage has 38047 configurations. Given 449 tasks for meta-training, Tensor-OBOE needs a tensor size $36950 \times 38047 \times 447=631 \times 10^9$.  If we consider every cell as a float occupying 4 bytes, this means roughly 157 GB for RAM.
>
> > 6. **Reviewer's Question:** *Minor Points*.
>
> 5. **Author's Answer:** We corrected these minor points in the revised version of the manuscript.

---

> > ### Comment · Reviewer_5R6t · 2022-11-22
> > **Thanks for the response**
> >
> > I would like to thank the authors for their detailed responses. The additional ablation study and baseline comparison experiments helped me to understand the paper better. A minor point is that stages addition or removal would also require the retraining of the aggregator, but I guess this scenario happens far less often. I have increased to score to 6.

---

### Official Review · Reviewer_KTJe · 2022-10-24

**Confidence:** 5
**Correctness:** 2
**Technical Novelty And Significance:** 3
**Empirical Novelty And Significance:** 2
**Recommendation:** 6

**Clarity, Quality, Novelty And Reproducibility:**

AutoML is a future goal for ML community. An expressive ML pipeline is useful for applications. In this topic, BO optimization is a typical way for parameter selection. The authors follow this framework and present their ideas under feature embedding. From this perspective, it likes a feature transformation study. It still belongs to the feature processing component in AutoML. It is thus the contribution is limited.

**Strength And Weaknesses:**

Questions:

(1) In Equation (1), $p^*$ and $\lambda(p^*)$  are the parameter to be optimized, and the author has set $p_{\lambda} = (p,\lambda(p))$, in section 4, the representation of neural network is $\phi\left(p_\lambda ; \theta\right): \operatorname{dom}\left(p_\lambda\right) \rightarrow \mathbb{R}^Z
$, but Equation (3) does not reflect that $p$ is encoded into MLP, please explain it.

(2) The optimization of $\theta=\left\{\theta^{\text {enc }}, \theta^{\text {aggr }}\right\}$ is confusing. Minimizing the negative log-likelihood of the GP requires a detailed explanation.

(3) The explanation of Equation (5) in Appendix G is too simple and needs a little description.

(4) In figure 1, the 'selector' step can be explained more clearly.

A lot of work has been done and compared with the previous work. From the experimental results, this neural network-based representation method has achieved great success.


**Summary Of The Paper:**

The selection of the algorithms and their hyperparameters is known as Pipeline Optimization. For most existing Pipeline Optimization, techniques do not explore the deep interaction between pipeline stages/components (e.g. between hyperparameters of the deployed preprocessing algorithm and the hyperparameters of a classifier).  This paper aims to capture the deep interaction between components of a Machine Learning pipeline and introduces DeepPipe, a neural network architecture for embedding pipeline configurations on a latent space. Such deep representations are combined with Gaussian Processes (GP) for tuning pipelines with Bayesian Optimization (BO). In experiments, this article demonstrates that the proposed pipeline representation helps to achieve state-of-the-art results in optimizing pipelines for fine-tuning deep computer vision networks.

**Summary Of The Review:**

The authors present some useful feature-processing ideas in AutoML. However, they didn't jump out of the constraint of the given AutoML framework. Moreover, the experimental results are not strong enough. There are only a few baselines, and some of them are too old. As far as I know, they are many BO works that can be compared.

---

> ### Author Response · Authors · 2022-11-16
> **Reply to Reviewer KTJe**
>
> Dear Reviewer,
>
> Thank you for your observations and remarks. We address each of them by referring to the new updated version of the paper:
>
> > 1. **Reviewer's Question:** *In section 4, the representation of neural network is $\phi(p_{\lambda};\theta): \mathrm{dom}(p_{\lambda}) \rightarrow \mathbb{R}^Z $, but Equation (3) does not reflect that is encoded into MLP, please explain it.*
>
> 1. **Author's Answer:** Thank you. We clarified this notation in the updated version (see updated Section 4.1). In the new notation, we differentiate clearly between the whole network $\phi$ and just the aggregation layer $\psi$.
>
> > 2. **Reviewer's Question:** *The optimization of $\theta=\left{\theta^{\text {enc }}, \theta^{\text {aggr }}\right}$ is confusing. Minimizing the negative log-likelihood of the GP requires a detailed explanation.*
>
> 2. **Author's Answer:** That is a valid point. We address this concern in the Appendix J of the updated manuscript.
>
> > 3. **Reviewer's Question:** *The explanation of Equation (5) in Appendix G is too simple and needs a little description.*
>
> 3. **Author's Answer:** We extended the previous Appendix G (now it is Appendix J).
>
> > 4. **Reviewer's Question:** *In figure 1, the 'selector' step can be explained more clearly. A lot of work has been done and compared with the previous work.*
>
> 4. **Author's Answer:** We address this concern in the new manuscript version. See Section 4.1 last paragraph. Basically, In the selector, we just pass through the output of the active encoder and ignore the outputs of the other encoders (they are non-active).
> Formally, within the $i$-th stage, only the output of one encoder is concatenated, therefore the output of the *Selector* corresponds to the active algorithm in the $i$-th stage and can be formalized as $\xi^{(i,p_i)}(\lambda_{i, p_i}) = \sum_{j=1}^{M_i} \mathbb{I}(j=a) \cdot \xi^{(i,j)}\left(\lambda_{i, j}\right)$, where $a$ is the index of the active algorithm, $M_i$ is the number of algorithms in the $i$-th stage and $\mathbb{I}$ denotes the indicator function.
>
> We hope that your main concerns are cleared. Otherwise, we will be happy to discuss and clarify further.

---

> > ### Comment · Reviewer_KTJe · 2022-11-23
> > **The novelty is unclear.**
> >
> > Thanks for your clarity. The paper proposes a pipeline that employs deep learning tools. The idea of adopting a  per-component encoder mechanism makes sense to me. However, my biggest concern is its novelty. From BO optimization, the contribution is not clear. In AutoML,  parameter optimization is a key step. So, I cannot increase my score as a "Standard Accept". The title and abstract need improvement to present the difference or idea/novelty. Subsection 3.2  does not give clear contributions on BO.
> >
> > This is a good topic/direction.  I appreciate the authors' effort. But I think that the manuscript needs significant improvement.

---

> > > ### Author Response · Authors · 2022-11-23
> > > **Thanks for your comment**
> > >
> > > We are glad to hear that the reviewer finds our idea of adopting a per-component encoder mechanism sensible.
> > >
> > > (I)
> > >
> > > We would like to understand your comment "From BO optimization, the contribution is not clear." better:
> > >
> > > To avoid a misunderstanding, BO is a policy for Hyperparameter Optimization that follows three stages:
> > >
> > > --
> > >
> > > repeat until budget time out:
> > >
> > > a) Fit a surrogate to approximate the observed validation accuracies of hyperparameter configurations
> > >
> > > b) Run an acquisition function to recommend the next configuration
> > >
> > > c) Evaluate the validation accuracy of the recommended hyperparameter configuration in b)
> > >
> > > --
> > >
> > > Our contribution in the domain of BO is in step a). We propose a novel surrogate to approximate the validation accuracy of pipelines. The proposed surrogate is novel as we introduce a new pipeline embedding for a deep-kernel Gaussian Process.
> > >
> > > Our novelty in the realm of BO is highlighted in the abstract:
> > >
> > > "... deep kernel Gaussian Process surrogates inside a Bayesian Optimization setup"
> > >
> > > as well as in the introduction
> > >
> > > "We introduce DeepPipe, a neural network architecture for embedding pipeline configurations on a latent space. Such deep representations are combined with Gaussian Processes (GP) for tuning pipelines with Bayesian Optimization (BO)."
> > >
> > > also, the first bullet point of the contributions makes our BO novelty explicit:
> > >
> > > "We introduce DeepPipe, a surrogate for BO that achieves peak performance when optimizing a pipeline for a new dataset through transfer learning."
> > >
> > > In addition, Section 3.2 describes BO briefly and Section 4 explains our novelty in terms of pipeline embeddings for deep-kernel GPs as surrogates.
> > >
> > > The embedding of pipelines is a novel neural architecture, and we demonstrated detailed empirical evidences on its usefulness compared to standard GPs, or deep-kernel GPs with a general embedding network.
> > >
> > > Do you still find the delineation of our novelty unclear? We would be glad to hear your suggestion on what can be clarified further.
> > >
> > > (II)
> > >
> > > Regarding the title and abstract edition:
> > >
> > > a) Following your recommendation, we can edit our abstract from the following:
> > >
> > > We propose embedding pipelines in a deep latent representation through a novel per-component encoder mechanism.
> > >
> > >
> > > to:
> > >
> > >
> > > We propose a novel Bayesian Optimization surrogate that embeds pipelines in a deep latent representation through a novel per-component encoder mechanism.
> > >
> > > b) Regarding the title what would be your recommendation?
> > >
> > > Would:
> > >
> > > Pipeline Optimization with Deep Pipeline Representations
> > >
> > > be an alternative?

---

> > > > ### Comment · Reviewer_KTJe · 2022-11-23
> > > > **Thanks for the response.**
> > > >
> > > > The steps are correct.
> > > >
> > > >  " a) Fit a surrogate to approximate the observed validation accuracies of hyperparameter configurations
> > > >
> > > > b) Run an acquisition function to recommend the next configuration
> > > >
> > > > c) Evaluate the validation accuracy of the recommended hyperparameter configuration in b)
> > > >
> > > > --"
> > > >
> > > > Our contribution in the domain of BO is in step a). We propose a novel surrogate to approximate the validation accuracy of pipelines. The proposed surrogate is novel as we introduce a new pipeline embedding for a deep-kernel Gaussian Process.
> > > >
> > > > There are many approaches to learn a surrogate function/distribution in ML. I am not sure whether your approach is better than typical ones. You define your way in your manner, but lack deep/solid reasons.
> > > > Could you present some reasons for "why it is better"?
> > > >
> > > > Indeed, as an ML researcher, I support ML ideas since there is too much deep learning work at each conference.
> > > > I support your work, but cannot learn more insights from the current presentation.  All scores are borderline, and make the reviewers very difficult to increase the score to accept.  I think, after careful revision, this draft would be much better.

---

> > > > > ### Author Response · Authors · 2022-11-23
> > > > > **Thanks for the interesting question**
> > > > >
> > > > > Thanks a lot for dedicating time to the rebuttal, and for your positive criticism which helps improve the quality of the manuscript. We appreciate your efforts.
> > > > >
> > > > > We can try to briefly answer your question "Could you present some reasons for "why it is better"?"
> > > > >
> > > > > Statement (i)
> > > > >
> > > > > Pipeline Optimization (PO) is a sub-problem of Hyperparameter Optimization (HPO). We assume the reader to be familiar with HPO. In the case of PO, the hyper-parameters have a specific structure, they are hierarchically structured stagewise, following the structure of Machine Learning pipelines.
> > > > >
> > > > > Statement (ii)
> > > > >
> > > > > In the field of HPO, the primary optimization algorithm is Bayesian Optimization (BO). It is based on surrogates that are probabilistic because the posterior uncertainty is used during the acquisition to explore regions of hyper-parameters with high uncertainty.
> > > > >
> > > > > Statement (iii)
> > > > >
> > > > > Within the BO literature, Gaussian Processes are the surrogate of choice. Their advantage is the ability to model uncertainty well with a limited number of evaluations. In HPO we can only evaluate a limited number of hyperparameter configurations, therefore, GPs have emerged as the most suitable surrogate family.
> > > > >
> > > > > Statement (iv)
> > > > >
> > > > > GPs are non-parametric models, therefore, have two limitations. On the one hand, without parameters, GPs cannot capture the deep interaction between input features. Secondly, due to being non-parametric GPs cannot be meta-learned. One way to resolve the limitation of GPs is to introduce a neural network embedding inside the GP kernel, known as Deep Kernel Learning [1]. Deep-kernel GPs were shown to be state-of-the-art in HPO [2], especially when the embedding is meta-learned.
> > > > >
> > > > >
> > > > > Now let us connect the statements:
> > > > >
> > > > > - Deep kernel GPs are the SOTA in HPO. (statement 4, background in statements 3-2).
> > > > > - Pipeline Optimization is a sub-problem of HPO (statement 1).
> > > > >
> > > > > Then how to extend the SOTA in HPO, to realm of PO (a sub-problem of HPO)?
> > > > >
> > > > > It seems very natural to us that the answer is:
> > > > > Adopt the SOTA in HPO towards PO, by taking into account the special characteristics of PO in contrast to HPO.
> > > > >
> > > > >
> > > > > As PO introduces a specific structure to the space of hyper-parameters, we propose an architecture for embedding the hyper-parameters that encapsulates the stage-wise design of pipelines.
> > > > >
> > > > >
> > > > > As a result, we believe there is a strong methodological motivation for the design choices of our method.
> > > > >
> > > > >
> > > > > Does that provide a clear answer to your question?
> > > > >
> > > > >
> > > > >
> > > > > [1] Wilson et al., ICML 2016, Deep Kernel Learning,
> > > > > [2] Wistuba, Grabocka, ICLR 2021, Few-Shot Bayesian Optimization with Deep Kernel Surrogates

---

### Official Review · Reviewer_2otF · 2022-10-24

**Confidence:** 4
**Correctness:** 4
**Technical Novelty And Significance:** 2
**Empirical Novelty And Significance:** 3
**Recommendation:** 6

**Clarity, Quality, Novelty And Reproducibility:**

Clarify: The paper is well written and mostly clear.

Quality: The paper doesn't include theoretical results, but it is high-quality as an empirical paper.

Novelty: The components used in the proposed algorithm are not really novel, but they are put together in a reasonable and novel way and combine to give a practical algorithm.

Reproducibility: Details about the algorithm and the experimental settings are included, and the code is uploaded.

**Strength And Weaknesses:**

Strengths:
- The idea of per-component encoders is particularly interesting since it allows the propose method to be extendable. And this extendability has been empirically verified.
- The experiments are extensive, and they do clearly show the empirical advantage of the proposed method. The way the experiments section (Section 5) is written makes it particularly easy to understand what questions/hypotheses the experiments are trying to answer.
- Last paragraph of Section 6, the insight provided in this paragraph is particularly nice.
- The paper is very well written, the proposed method is well explained.

Weaknesses:
- Section 5.4, top of page 8: It is worrying that different values of F and numbers of encoder layers are used in different experiments. How are these parameters chosen? Ideally, the algorithm should work well in different experiments without requiring the tuning of these parameters; on the other hand, if these parameters are tuned for every experiment, this will be unrealistic since we usually don't get to do this in practice when faced with a new task. This is my biggest concern, please clarify.
- The paper claims that the proposed method is able to take into account the interactions among different components, but the paper never made it clear which specific algorithmic design(s) has made this possible. I think this is an important insight and should be explicitly discussed.
- Last paragraph of Section 6, the insight provided in this paragraph is nice as I said above, but why have you used randomly initialized weights for DeepPipe rather than using the weights after training?
- Section 5.3, Hypothesis 5: I wonder what's the implication of this hypothesis? In other words, what insights can we draw about your algorithm if this hypothesis is validated (which it is)?
- Page 8, last paragraph: the presentation of this paragraph can be improved, it's difficult to understand in the current form.
- [minor] Section 3.2, equation (2): it looks like you are assuming noiseless observations, which is different from what is written in the paragraph above where the observations are noisy. Perhaps the paragraph should be modified to discuss noiseless observations.
- [minor] Section 4.2, equation (5): should give some intuitions as to how this objective is designed.
- [minor] Section 5.4: The notations in this section are kind of inconsistent with the previous ones such as those in Section 4.1. Is it true that  $L_i=F(Q_i+M_i)$?
- [minor] Last line of page 7: I think the left bracket is in the wrong place, I think it should be $F\sum_i(Q_i+M_i)$

**Summary Of The Paper:**

This paper proposes a novel approach to pipeline optimization for machine learning models. The proposed method encodes the hyperparameters of every algorithm into a latent space and then aggregates the hyperparameters of the algorithms from all stages into a single latent representation. Based on this learned latent representation, the algorithm uses Bayesian optimization with deep kernel learning to optimize the pipelines, and also allows meta-learning using existing datasets. Extensive experiments are performed to show the practical efficacy of the propose method.

**Summary Of The Review:**

The method proposed in this paper is intuitive and useful, and the experiments are comprehensive and nicely done. My only major concern is the first comment I listed above under "Weaknesses" regarding whether some of the algorithmic hyperaprameters have been unfairly fine-tuned for different experiments.

---

> ### Author Response · Authors · 2022-11-16
> **Reply to Reviewer 2otF**
>
> Thank you for your observations and suggestions. We address each of them by referring to the new updated version of the paper:
>
>  >1. **Reviewer's Question:** *Section 5.4, top of page 8: It is worrying that different values of F and numbers of encoder layers are used in different experiments. How are these parameters chosen? Ideally, the algorithm should work well in different experiments without requiring the tuning of these parameters; on the other hand, if these parameters are tuned for every experiment, this will be unrealistic since we usually don't get to do this in practice when faced with a new task. This is my biggest concern, please clarify.*
>
> 1. **Author's Answer**: F was chosen using the meta-validation dataset (with a small grid search {4,6,8,10}). We did this because the performance depends on the number of meta-training tasks and F (Appendix F). It depends on F because this factor and the input dimensionality of the search space control the expressivity of the network (section 5.4). Consequently, the value of F should be tuned for a new meta-dataset, when we have different meta-training tasks and input dimensions. Notice, we also tuned the HPs of baselines as described in Section 5.2.
>
>  >2. **Reviewer's Question:** *The paper claims that the proposed method is able to take into account the interactions among different components, but the paper never made it clear which specific algorithmic design(s) has made this possible. I think this is an important insight and should be explicitly discussed.*
>
> 2. **Author's Answer**: It is indeed a very important point. We added a new section (Appendix D) for discussing how this interaction is made possible.
>
> >3. **Reviewer's Question:** *Last paragraph of Section 6, the insight provided in this paragraph is nice as I said above, but why have you used randomly initialized weights for DeepPipe rather than using the weights after training?*
>
> 3. **Author's Answer**: Multiple prior studies use randomly initialized weights [1,2,3] to study the effect of the inductive bias introduced by the architecture in convolutional neural networks. So we also study the inductive bias of our architecture along these established lines. When you train from data it is more difficult to disentangle the prior assumptions induced solely by the architecture, from the effect of the data-generating process.
>
> [1] Wilson, Andrew G., and Pavel Izmailov. "Bayesian deep learning and a probabilistic perspective of generalization." Advances in neural information processing systems 33 (2020): 4697-4708.
>
> [2] Ulyanov, Dmitry, Andrea Vedaldi, and Victor Lempitsky. "Deep image prior." Proceedings of the IEEE conference on computer vision and pattern recognition. 2018.
>
> [3] Zhang, Chiyuan, et al. "Understanding deep learning (still) requires rethinking generalization." Communications of the ACM 64.3 (2021): 107-115.
>
> >4. **Reviewer's Question:** *Section 5.3, Hypothesis 5: I wonder what's the implication of this hypothesis? In other words, what insights can we draw about your algorithm if this hypothesis is validated (which it is)?*
>
> 4. **Author's Answer**: The implication is that if we have few or no meta-dataset for pre-training DeepPipe, we can rely on the assumption of the pipeline's performance introduced by the encoders (inductive bias). If we have enough meta-data, we do not need to introduce this assumption, because DeepPipe is able to meta-learn this (Appendix F).
>
> >5. **Reviewer's Question:** *Page 8, last paragraph: the presentation of this paragraph can be improved, it's difficult to understand in the current form.*
>
> 5. **Author's Answer**: Thank you. We have taken this into account in the revised version of the manuscript.
>
> >6. **Reviewer's Question:** *[minor] Section 3.2, equation (2): it looks like you are assuming noiseless observations, which is different from what is written in the paragraph above where the observations are noisy. Perhaps the paragraph should be modified to discuss noiseless observations.*
>
> 6. **Author's Answer**: You are right. We added a clarification on the noisy covariance matrix (see Section 3.2).
>
> >7. **Reviewer's Question:** *[minor] Section 4.2, equation (5): should give some intuitions as to how this objective is designed.*
>
> 7. **Author's Answer**: Thank you. We now incorporate information about the design of the loss function. See Section 4.2 and Appendix J.
>
> >8. **Reviewer's Question:** *[minor] Section 5.4: The notations in this section are kind of inconsistent with the previous ones such as those in Section 4.1.*
>
> 8. **Author's Answer**: Answer: Thank you. We modified the notation to make it consistent.
>
> >9. **Reviewer's Question:** *[minor] Last line of page 7: I think the left bracket is in the wrong place.*
>
> 9. **Author's Answer**: See point above.

---

> > ### Comment · Reviewer_2otF · 2022-11-23
> > **Thanks for the Response**
> >
> > I'd like to thank the authors for the response and the additional results. The response to my first question makes a lot of sense, using the meta-validation set to tune algorithmic hyperparameters is reasonable, and the explanation as to why F should be tuned for different tasks is also valid. The additional Appendix D is also a nice addition to the paper.
> >
> > I understand that the other reviewers have concerns regarding the novelty of the proposed method compared to previous related approaches, which is also a valid point and makes me hesitate as to whether it's appropriate to further increase the score to 8. But I do see the empirical value of the paper since it has used extensive experiments to demonstrate that the proposed natural and intuitive method works well for pipeline search in ML, and hence in addition to its own empirical contribution, can also serve as a nice baseline for future works in this area.

---

> > > ### Author Response · Authors · 2022-11-23
> > > **Thanks for the positive feedback**
> > >
> > > We appreciate your support.
> > >
> > > Regarding novelty, we understand the criticism at this point. Our method is a new component (surrogate) in an existing optimization technique (BO), and not a new optimization method.
> > >
> > > Indeed, our method relies heavily on the skeleton of Bayesian Optimization. Our novelty lies in proposing a new architecture for the embeddings of a BO surrogate.
> > >
> > > However, we do not share the opinion that proposing a new architecture is a minor contribution.
> > >
> > > For instance, Convolutional Neural Networks are a different architecture from Multilayer perceptrons. Residual Networks are an improved architecture to ConvNets. Transformers are simply a better architecture than ConvNets.
> > >
> > > If our method is based on BO (the optimization method), then ConvNets, ResidualNets, Transformers, etc. are based on supervised learning with stochastic gradient descent (the optimization method).
> > >
> > > Does designing new architectures, by enforcing specific neural connectivity patterns to match the prior knowledge of the problem, consist of a minor/negligible contribution?
> > > We believe the answer is: No if it brings significant gains in predictive performance.
> > >
> > > In our case, we empirically demonstrated that our architecture gives a significant lift in performance in the realm of Pipeline Optimization.

---

### Author Response · Authors · 2022-11-16
**General Reply**

We thank all the reviewers for their feedback and suggestions.

Based on the received comments, we conducted the following new experiments:


- **Additional Experiment 1 (Answer to Reviewers KTJe,5R6t)**: Comparison to AutoPrognosis [1], another SOTA method in Pipeline Optimization. We compared DeepPipe with AutoPrognosis and measured the computation time. We found that DeepPipe performs better than AutoPrognosis and runs 100 BO iterations 10 times faster than AutoPrognosis. See details in Appendix E or link: https://anonymous.4open.science/r/DeepPipe-3DDF/Comparison_with_AutoPrognosis.md

- **Additional Experiment 2 (Answer to Reviewer 2otF, 5R6t)**: Analyze/ablate the interactions of the latent encoder/aggregation layers. We run a meta-trained DeepPipe with and without encoder layers/aggregator layers. We find out that it is important to have encoder and aggregation layers, instead of using the raw input space or using just an encoder layer. Thus, it is important to capture interaction among hyperparameters in a stage (via the encoder) and among stages (via the aggregator). See appendix D or link: https://anonymous.4open.science/r/DeepPipe-3DDF/Discussion_on_Interactions.md


- **Additional Experiment 3 (Answer to Reviewer 5R6t)**: Investigate the number of evaluated pipelines per wallclock time, in comparison to the baselines. We count the number of pipelines observed by the methods in Experiment 3 on OpenML datasets. We found out that DeepPipe explores a similar amount of pipelines to other methods, which indicates that the optimization overhead introduced by our methods is not prohibitive. See Appendix C or link: https://anonymous.4open.science/r/DeepPipe-3DDF/Number_of_Pipelines.md


In addition, we updated the main manuscript with the following changes:

- Clarification on the noisy covariance matrix (see Section 3.2).
- Clarification on the notation for the aggregation layer (see Section 4.1).
- Clarification on the selector module (see Section 4.1 last paragraph).
- Extensions and clarifications about the loss function and the meta-training (see Section 4.2 and Appendix J).
- Matching color for Random Search (RS) plots (Figures 2 and 3).
- Clarification on the discussion for the results of Experiment 4 (second but last paragraph, Section 6).

The changes in the manuscript are highlighted in blue.
We hope that the main concerns of the reviewers are cleared. Otherwise, we will be happy to discuss and clarify further.

References:

[1] Alaa, Ahmed, and Mihaela Schaar. "Autoprognosis: Automated clinical prognostic modeling via bayesian optimization with structured kernel learning." International conference on machine learning. PMLR, 2018.

---

### Decision · Program_Chairs · 2023-01-20

**Decision:**

Reject

**Justification For Why Not Higher Score:**

The main issue with this work is lack of novelty. This lack could be overcome by empirical results on harder AutoML problems with limited transfer from closely related tasks, where the assumed modularity here could make a bigger difference.

It should be said that the authors promise a number of additional experiments, but without the context of a surrounding paper, these remain too hard to check for the reviewers to change their vote and champion this paper.


**Justification For Why Not Lower Score:**

N/A

**Metareview: Summary, Strengths And Weaknesses:**

This paper tackles joint tuning of stages in a preprocessing and feature pipeline along with search over algorithm and hyperparameters. This is achieved by learning neural encodings of components of the pipeline. Due to the modularity of the setup, the authors claim this an be done even from limited transfer data. They then use deep kernel and GP-based BO in order to perform the sequential tuning.

All reviewers acknowledge the practical relevance of this work, as joint optimization over pipelines and hyperparameters still does not have a very convincing solution, at least in practice. However, the present work remains rather close to prior work, both on deep kernels in the transfer BO context, and neural encodings of components (possibly more shallow), which implies a lack of novelty.

The authors are encouraged to apply their work to more challenging AutoML problems, where the suggested advantages of their proposal may make a clear difference. While they proposed extra results in the rebuttal, their relevance could not be checked by the reviewers (lack of context). They may focus in particular on robustness to limited transfer data, maybe of not closely related tasks, and on making their approach fully automatic (it still needs tuning of parameters using some data for each new task).